# Multi-target and ultra-high-speed optical wireless communication using a thin-film lithium niobate optical phased array

Xiaoyue Ma[1,5], Mingrui Yuan [1,5], Jingchi Li[2], Hongdong Zhang[1], Baichuan He[1], Pu Zhang[1], Yongheng Jiang[1], Huifu Xiao[1], Guanghui Ren [3], Arnan Mitchell [3], Yikai Su [2] & Yonghui Tian [1,4] ✉

Optical wireless communication (OWC) effectively addresses challenges such as radio spectrum scarcity and signal attenuation by leveraging the properties of laser beams. A major advance in this field comes from the incorporation of optical phased arrays (OPAs), which enable inertial-free, high-speed beam steering with transformative potential. In this contribution, we propose and demonstrate a multi-target and ultra-high-speed OWC system based on a thin-film lithium niobate (TFLN) OPA. It enables real-time multi-target connection without mechanical components or lenses. This system can achieve OWC with a single-channel communication data rate of up to 320 Gbps in the modulation format of 16-Quadrature Amplitude Modulation (QAM), significantly exceeding the peak capabilities of 5 G and current 6 G proposals. System performance is further validated through the stable transmission of uncompressed high-definition video. This work establishes a new paradigm for fully solid-state, chip-scale OWC systems, combining unprecedented single-channel data throughput with dynamic multi-target support.

Optical wireless communication (OWC) leverages the high mono-chromaticity, coherence, and directivity of laser sources to establish high-capacity, low-latency directional links with excellent anti-interference capabilities. It thus offers a promising solution to current challenges in modern communications, including radio frequency spectrum scarcity and signal attenuation[1–3]. Conventional OWC systems typically rely on point-to-point architectures, which are unable to support multicast or broadcast services from a single source to multiple users. Dynamic beam steering has emerged as a viable approach to overcome this limitation[4–6]. This technique provides inertia-free operation, sub-millisecond response times, and high programmability, significantly enhancing the versatility and practicality of OWC systems. These improvements show considerable potential for demanding applications, including mobile access

networks, satellite communications, and emergency response systems[7–10]. Current dynamic beam-steering techniques mainly involve three approaches: metasurfaces[11,12], mechanical beam-steering systems[13–16], and optical phased arrays (OPAs)[17–21]. Metasurfaces employ artificially engineered subwavelength structures to achieve beam deflection, focusing, and mode conversion[11,12]. However, the optical responses of most metasurface devices are static once fabricated. Even those with thermal tuning often suffer from slow response speeds and limited tuning ranges. Mechanical beam-steering systems adjust the beam direction by moving optical components or transceiver modules, yet these systems often compromise reliability and integration potential[13–16]. In contrast, fully solid-state, miniaturized, and high-speed dynamic beam steering is a pivotal direction for next-generation OWC systems. OPAs enable

¹School of Physical Science and Technology, Lanzhou University, Lanzhou, Gansu, China. ²State Key Laboratory of Photonics and Communications, School of Information and Electronic Engineering, Shanghai Jiao Tong University, Shanghai, China. ³Integrated Photonics and Applications Centre (InPAC), School of Engineering, RMIT University, Melbourne, VIC, Australia. ⁴School of Mathematics and Physics, North China Electric Power University, Beijing, China. ⁵These authors contributed equally: Xiaoyue Ma, Mingrui Yuan. ✉e-mail: tianyh@lzu.edu.cn

non-mechanical beam steering by precisely controlling the phase of individual optical antenna elements, making them a key technology in this field[17–21].

OPAs have been successfully implemented on several integrated photonic platforms, including silicon[22–27], silicon nitride ($Si_3N_4$)[28–30], thin-film lithium niobate (TFLN)[31–34], and heterogeneous integrations of $Si_3N_4$ and TFLN[35,36]. Silicon-based OPAs benefit from complementary metal-oxide-semiconductor (CMOS) compatibility and high integration density. However, their thermo-optic phase shifters exhibit microsecond-scale response times and require continuous power consumption to maintain thermal stability. While $Si_3N_4$ platforms provide low propagation loss and broad optical transparency, the lack of a linear electro-optic effect limits high-speed modulation. In contrast, TFLN has emerged as a leading platform for high-performance OPAs[37–40], owing to its low loss, high stability, wide transparency window, and strong Pockels effect. Furthermore, TFLN is compatible with cost-effective wafer-scale fabrication, supporting a viable path toward mass commercialization. Recent progress in TFLN-based OPAs demonstrates notable capabilities. Passive OPAs achieve wide beam steering via field-of-view (FOV) stitching, though their lack of dynamic programmability limits applicability in multi-target communication scenarios[34]. Active OPAs face inherent design trade-offs: larger antenna spacing improves the beam full width at half maximum (FWHM) but reduces the FOV[31]. Sparse arrays can narrow beam divergence and enhance pointing accuracy, albeit at the cost of limited angular coverage[32,33]. Additionally, a low sidelobe suppression ratio (SLSR) degrades the signal-to-noise ratio at the receiver, increasing the bit error rate (BER) in OWC links. On hybrid platforms combining $Si_3N_4$ and TFLN, OPAs using Taylor amplitude distributions achieve improved SLSR, but at the cost of longer beam-steering times. This results in communication latency and reduced real-time performance[36]. Despite these device-level advances, system-level demonstrations of TFLN-based OPAs for high-speed OWC remain limited. Therefore, integrating TFLN OPAs into system-level OWC architectures represents a transformative opportunity.

In this contribution, we propose and demonstrate a multi-target ultra-high-speed OWC system utilizing a TFLN OPA. The system incorporates an optimized curved coupled trapezoidal grating antenna that enables high-resolution two-dimensional beam steering across a 62° × 11° FOV via independent phase and wavelength control. An SLSR as low as −13.6 dB is achieved, facilitating multi-target tracking and interference suppression. By independently manipulating the transverse phase and longitudinal wavelength dimensions, a 4 × 4 point-cloud matrix is generated within a 40° × 10° FOV, successfully reconstructing the patterns of the letters "L", "Z", and "U". Leveraging the strong Pockels effect in TFLN, the system demonstrates rapid beam steering with a record-breaking transition time of approximately 158 ps and a higher modulation efficiency of 8.73 pJ/π, compared to these reported TFLN-based OPAs. Our proposed system enables real-time multi-target communication without mechanical moving parts or external lenses. Experimental results confirm high beam direction accuracy and stability, indicating the system's suitability for optical wireless links. Notably, OWC with a single-channel communication data rate of up to 320 Gbps is achieved using 16-Quadrature Amplitude Modulation (QAM) format at six distinct angles within the FOV of 31° × 3.2°, significantly exceeding previous records and surpassing the peak rates of 5G and current 6G proposals. System reliability is further validated through the stable transmission of uncompressed high-definition (HD) video. This work establishes a fully solid-state chip-scale OWC system that combines ultra-high throughput with dynamic multi-target capability, thereby providing a foundation for next-generation satellite networks, secure drone communications, and immersive metaverse connectivity.

## Results
### Principles and designs

Figure 1a illustrates the working principle of an OPA-based OWC system. Information is transmitted from a data center via the OPA, which emits optical beams at specific steering angles. Each beam carries an independent data stream directed toward a receiving center. These distinct streams are subsequently distributed to end users and captured by dedicated receivers. A significant feature of the OPA is its wide FOV and high directivity, allowing each beam to establish a separate communication link. The system supports multi-target access through high-speed beam steering, while also improving channel confidentiality and security. Figure 1b presents the schematic of the core OPA device used in the OWC system. The device is fabricated on an *X*-cut TFLN platform, with a footprint of 10 mm × 1.1 mm. It consists of a 600-nm-thick top lithium niobate layer and a 2-μm-thick buried oxide layer. The rib waveguides are etched with a sidewall angle of approximately 67°, a width of 1 μm, a slab thickness of 300 nm, and a rib height of 300 nm (see Supplementary Note 1). Key components integrated on the chip include multimode interference (MMI) coupler trees, a voltage-controlled phase shifter array, uniformly spaced curved waveguides, and an optical antenna array. Input light at around 1550 nm is coupled into the chip via edge couplers. The MMI coupler tree splits the optical signal uniformly into 16 channels. Each channel contains an electro-optic phase shifter that can be independently adjusted to provide phase shifts from 0 to 2π. The output waveguides from the phase shifters are routed through curved sections and compressed to a pitch of 1.5 μm before connecting to trapezoidal slab antennas. The radii of adjacent curved waveguides are optimized to reduce mutual coupling. Light from each waveguide is coupled into a trapezoidal antenna and radiated into free space. The compact design and high level of integration facilitate practical deployment in OWC systems, while the optimized waveguide geometry and phase control ensure efficient and stable optical signal transmission. To validate the fabrication accuracy of the key waveguide structures in the OPA chip, Fig. 1c presents a scanning electron microscopy (SEM) image of the rib waveguide cross-section, which exhibits a well-defined geometry consistent with the designed dimensions. The OPA chip utilizes co-packaged optics, and Fig. 1d shows a micrograph of the fully assembled 16-channel device. The chip is wire-bonded to a printed circuit board (PCB), as illustrated in Fig. 1e, to enable electrical interfacing with external drive circuits for voltage application to the phase shifters and to facilitate beam control during experiments.

To address the need for flexible and efficient beam control in OWC systems, we introduce an OPA capable of independent beam steering in two dimensions. Transverse steering, controlled via phase modulation, is achieved through electro-optic phase shifters, while longitudinal steering, governed by wavelength adjustment, is facilitated using tunable lasers. In the transverse dimension, the maximum steering angle is limited by the antenna spacing. Reducing the inter-antenna spacing to half the operating wavelength permits beam steering of up to 180°. However, such narrow spacing increases inter-element coupling and crosstalk, as discussed in Supplementary Note 2. Conventional strategies often utilize a non-uniform antenna array to suppress inter-waveguide crosstalk, thereby improving the resolution of OPAs. However, achieving a wide FOV generally requires the incorporation of specialized structures fabricated through complex techniques to achieve a flattened far-field profile. To address these issues, we design and optimize a trapezoidal antenna structure[22]. This configuration effectively suppresses edge scattering and improves the SLSR. Additional details are provided in Supplementary Note 3. Combined with a curved coupling scheme, the trapezoidal antenna architecture enables reduced waveguide spacing and crosstalk, allowing antennas to be spaced at intervals approaching half the wavelength. Consequently, the beam steering range in the transverse dimension is substantially enhanced (see Supplementary Note 4). For the

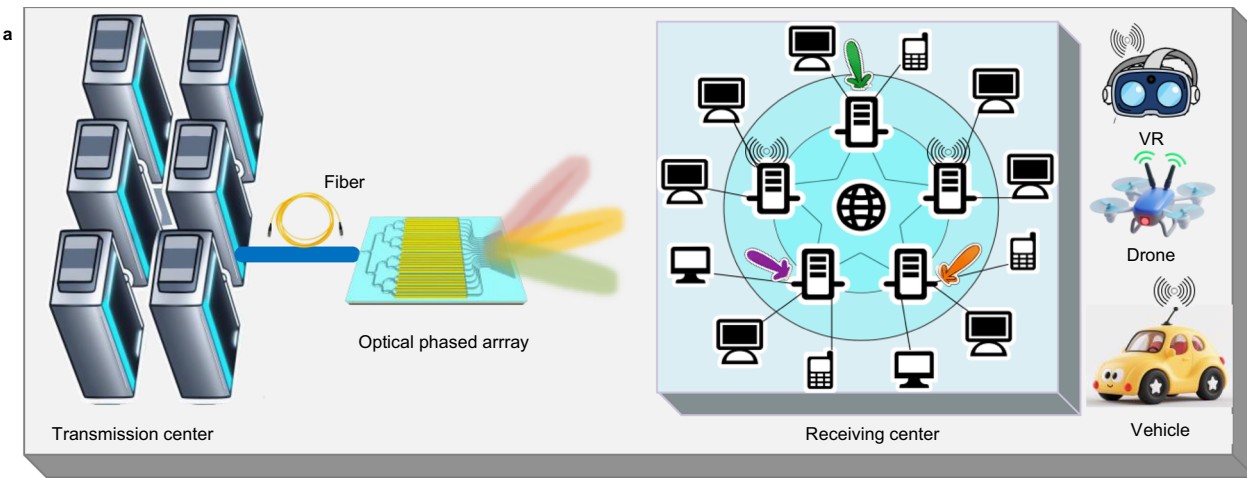

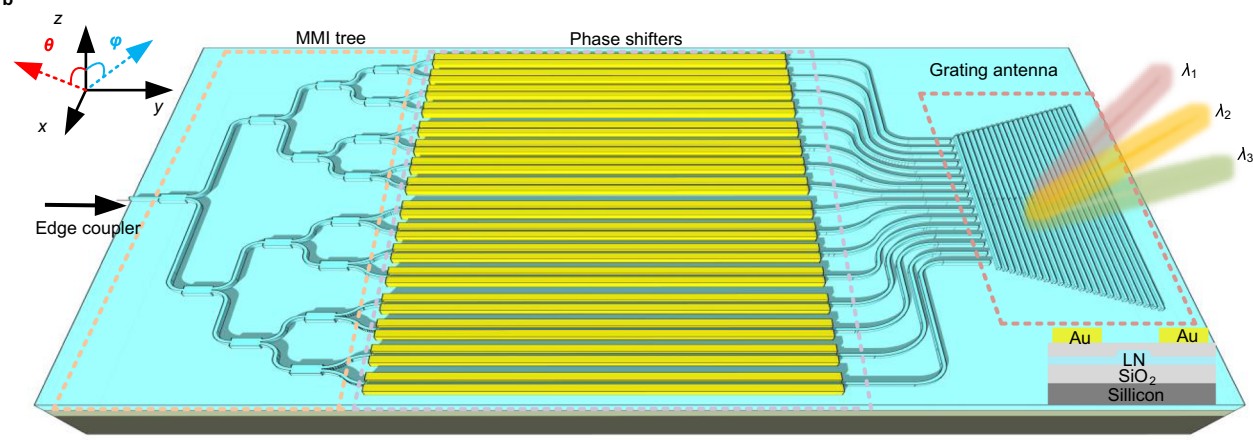

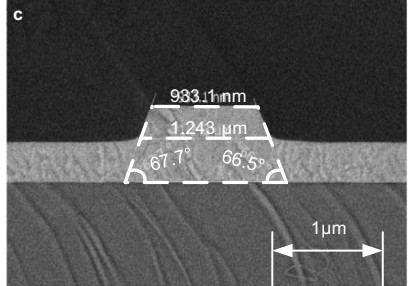
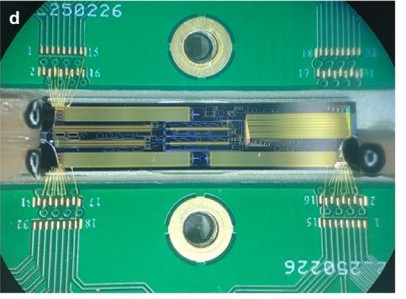
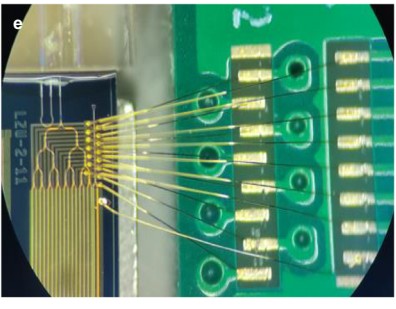

**Fig. 1 | Design of the TFLN-based OPA. a** Schematic illustration of the operational scenario for OWC using the OPA. Data centers transmit independent data streams through optical fibers to the OPA. The OPA emits light beams at specific steering angles to deliver these data streams to the receiving center. The receiving center then distributes the distinct data streams to end users. **b** Schematic diagram depicting the OPA structure. **c** SEM image of the rib waveguide cross-section. **d** Microscopic image of the packaged chip. **e** Microscopic image of the wire bonding electrode region. MMI multimode interference, LN lithium niobate.

longitudinal dimension, the steering range depends primarily on two factors: the wavelength tuning range of the laser and the operational bandwidth of the passive components. Detailed specifications are provided in Supplementary Note 4.

To assess the stability of transverse beam steering in the proposed OPA at a fixed wavelength, Fig. 2a presents the far-field beam spots at transverse steering angles of ±31°, ±24°, ±18°, ±12°, ±6°, and 0°. These measurements are conducted at a wavelength of 1550 nm. The beam profiles remain distinct and well-defined across all angles, confirming that the OPA maintains consistent transverse steering performance without significant degradation. This stability is critical for reliable signal transmission in OWC systems, as it minimizes beam quality deterioration during angular adjustments. To investigate the influence of transverse steering angles on the optical field distribution at various wavelengths within the C-band, Fig. 2b displays the normalized optical

field distributions for several steering angles at wavelengths of 1530 nm, 1550 nm, and 1565 nm. The beam is calibrated using a gradient descent algorithm (see Supplementary Note 5). The optical field distributions exhibit uniform profiles across all wavelength and angle combinations, indicating that the calibration process effectively compensates for variations induced by fabrication imperfections and other disturbances. This consistency underscores the suitability of the OPA for multi-wavelength OWC applications, where stable field distribution is essential for maintaining signal integrity. Figure 2c characterizes the FWHM of the far-field beam profile, a key parameter for evaluating beam collimation. The measured FWHM values are 3.8° in the transverse direction ($\theta$) and 1.72° in the longitudinal direction ($\varphi$). The inset depicts the corresponding far-field beam profile after phase calibration. The narrow FWHM values demonstrate the capability of the OPA to generate highly collimated beams, which reduces signal

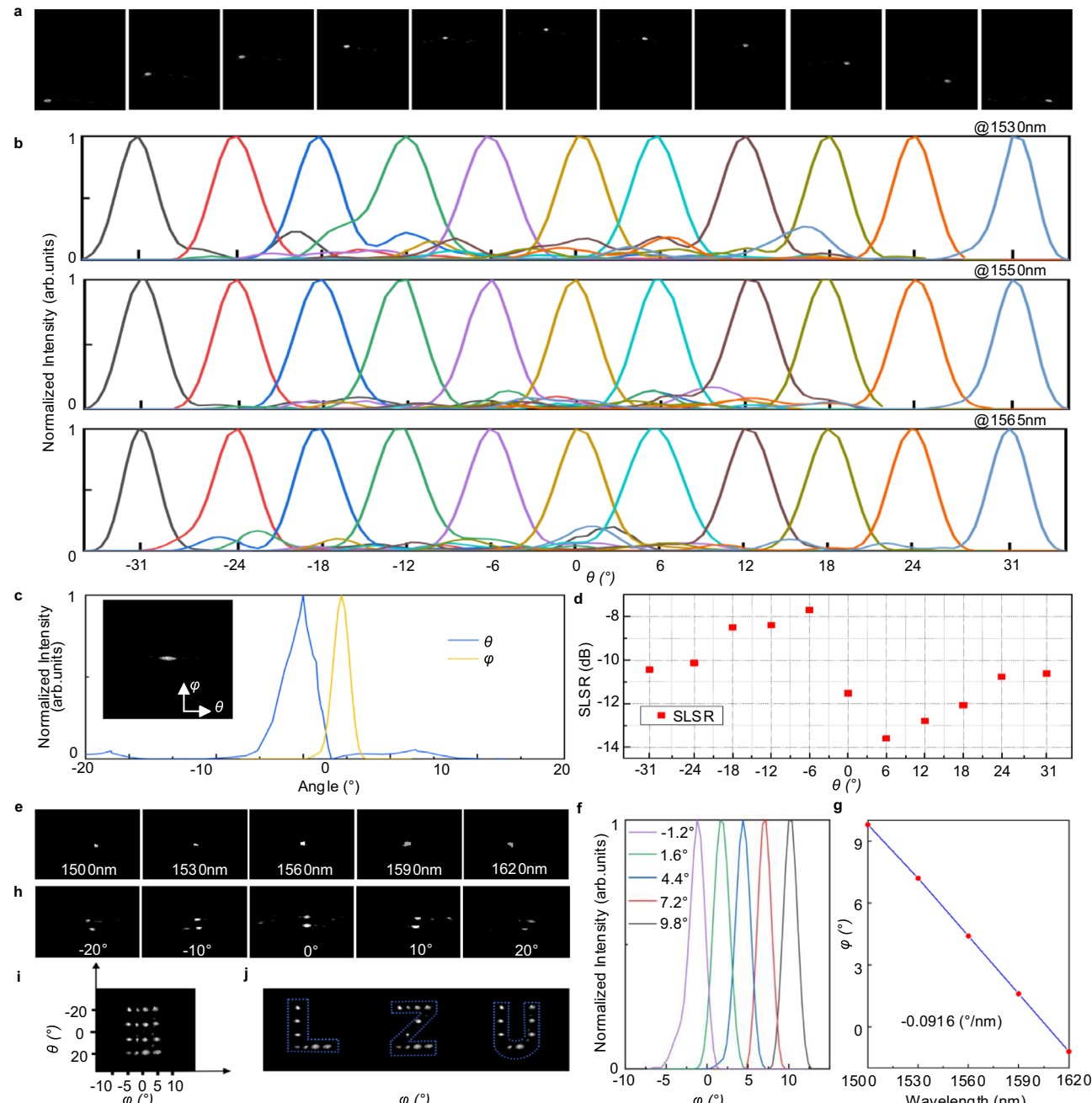

**Fig. 2 | Beam steering performance characterization of the OPA. a** Far-field beam spots measured at transverse steering angles of ±31°, ±24°, ±18°, ±12°, ±6°, and 0° at a wavelength of 1550 nm. **b** Normalized optical field distributions corresponding to different transverse steering angles at wavelengths of 1530 nm, 1550 nm, and 1565 nm. **c** Resolution of the measured far-field intensity profiles along the transverse and longitudinal dimensions at a phase angle of 0° and a wavelength of 1550 nm. The inset shows the corresponding far-field image after phase calibration. **d** The SLSR for various transverse steering angles at a wavelength of 1550 nm. **e** Far-field beam spot at wavelengths of 1500 nm, 1530 nm, 1560 nm, 1590 nm, 1620 nm. **f** Normalized optical field distributions measured at wavelengths of 1500 nm, 1530 nm, 1560 nm, 1590 nm, and 1620 nm. **g** Linear fitting of beam-steering angle as a function of wavelength. **h** Far-field beam spots measured at wavelengths of 1530 nm and 1630 nm for transverse steering angles of ±20°, ±10°, and 0°. **i** A 4 × 4 point-cloud matrix. **j** Projected characters "L", "Z", and "U" formed via beam steering. SLSR sidelobe suppression ratio.

divergence and enhances energy concentration in OWC links. The calibrated beam profile further confirms that phase correction significantly improves beam quality by reducing phase-related distortions. To evaluate the SLSR during transverse beam steering, a parameter pertinent to crosstalk in multi-beam systems, Fig. 2d shows the SLSR across a range of transverse steering angles. The SLSR reaches a peak value of −13.6 dB and remains below −7.8 dB throughout the angular range tested. This consistently low SLSR indicates effective suppression of sidelobe energy, thereby reducing

interference among adjacent beams. In multi-target OWC systems, this characteristic is vital for minimizing signal crosstalk and ensuring communication reliability.

To validate the longitudinal beam-steering performance of the OPA via wavelength tuning, Fig. 2e displays far-field beam spots at wavelengths of 1500 nm, 1530 nm, 1560 nm, 1590 nm, and 1620 nm. The beam spots shift monotonically with increasing wavelength, demonstrating clear longitudinal steering. Complementing the steering analysis, Fig. 2f shows the corresponding normalized optical field

distributions at the same wavelengths. The profiles maintain a consistent Gaussian shape under wavelength variation, aligning with the stable beam spots observed in Fig. 2e. This uniformity ensures that the optical field quality remains unaffected during steering, which is essential for preserving signal integrity across the operating range. For quantitative analysis of wavelength-dependent steering efficiency, Fig. 2g presents a linear fit of beam-steering angle versus wavelength. Over a 120 nm bandwidth, the OPA achieves a steering range of 11°. The fit yields a steering efficiency of −0.091°/nm. The linear relationship between wavelength and steering angle simplifies control and enables precise beam positioning via laser tuning. To verify the multi-target connectivity of the OPA, two laser beams at 1530 nm and 1630 nm are simultaneously injected into the device. The appearance of two distinct far-field spots on the screen confirms the OPA's ability to generate stable beams for multi-target operation. Figure 2h shows far-field beam spots at wavelengths of 1530 nm and 1630 nm for transverse steering angles of ±20°, ±10°, and 0°. The spots retain well-defined shapes across all angles, demonstrating that the OPA maintains stable beam quality while facilitating multi-target connectivity. To further demonstrate its practical utility, the OPA generates a 4 × 4 point-cloud matrix across a 40° × 10° FOV, as shown in Fig. 2i. This matrix is configured to form the characters "L", "Z", and "U", shown in Fig. 2j. This ability to project user-defined patterns confirms independent control over both transverse and longitudinal steering dimensions, supporting applications such as multi-target free-space communication or structured light imaging. Additional details on OPA performance are provided in Supplementary Note 6. Notably, we report two distinct FOV values for the same OPA device. The maximum device-level FOV of 62° × 11° is characterized from far-field scans under combined phase and wavelength control. The demonstration of pattern reconstruction is conducted within a 40° × 10° FOV, which is selected as it provides the necessary angular area to project a clearly defined 4 × 4 point-cloud pattern.

## Performance characterization of OPA

The half-wave voltage ($V_\pi$) is a key parameter for evaluating the performance of OPA phase shifters, as it dictates the driving voltage required to achieve a π-phase shift. This parameter is measured using the experimental setup shown in Fig. 3a. Through a polarization controller (PC), a 1550 nm laser source is coupled into a Mach–Zehnder interferometer (MZI) modulator, which incorporates two target OPA phase shifters. One arm of the MZI modulator is driven by a 10 MHz triangular wave signal generated by an arbitrary function generator (AFG). The modulated optical signal is detected by a photodetector (PD), and the PD output is recorded using an oscilloscope. Figure 3b plots the measured normalized optical intensity versus driving voltage. For the OPA phase shifter with an 8 mm electrode length, the measured $V_\pi$ is 4.7 V. The corresponding power consumption, calculated based on this $V_\pi$, is 8.73 pJ/π. Detailed derivations of the power consumption are provided in Supplementary Note 7. A $V_\pi$ of 4.7 V is beneficial for practical OPA deployments, as it reduces the demand for high-voltage drive electronics. Furthermore, the low power consumption of 8.73 pJ/π contributes to energy-efficient operation of OPA systems, which is an essential characteristic for portable and power-sensitive optical communication devices.

Steering speed represents another critical performance metric for OPA phase shifters, as it directly determines the dynamic response of the overall OPA. A higher steering speed enables faster adjustments of the beam direction, which is essential for dynamic OWC applications. To evaluate this parameter, the rise and fall times of the MZI modulator are measured using the setup illustrated in Fig. 3c. In this experiment, a 1550 nm laser source is employed, and light is coupled into the MZI modulator via a PC to maintain a stable polarization state. One arm of the modulator is driven by a 1 GHz square-wave voltage signal generated by an arbitrary waveform generator (AWG). The time-domain

response of the optical output is recorded using a digital communication analyzer (DCA). Figure 3d presents the waveform acquired from the time-domain response measurement in Fig. 3c. From this waveform, the rise and fall times of the MZI modulator are determined to be 170 ps and 146 ps, respectively. These short transition times reflect a fast dynamic response of the OPA phase shifter. In high-speed OWC systems, such as those used for real-time dynamic beam tracking or multi-target switching, such rapid responsiveness ensures that the OPA can adapt promptly to varying communication requirements, thereby enhancing the stability and reliability of the optical link.

The OPA can function as either a transmitter or a receiver owing to the reciprocity principle. To characterize its transmission performance across different distances, the OPA chip is employed as the transmitter, and a collimator serves as the receiver. Detailed experimental procedures are provided in Supplementary Note 8. Figure 3e presents the total link efficiency as a function of transmission distance, revealing a decline in efficiency with increasing distance. A linear fit yields an attenuation rate ($k$) of approximately −1.48 dB/m. In practical implementations, this attenuation can be compensated for by achieving more precise beam divergence control at the OPA output. The beam directivity of the OPA under wavelength shifts is also evaluated. Measurements indicate a 3 dB attenuation in power with a wavelength shift of only 2.2 nm, corresponding to an angular deviation of approximately ±0.1°. These results affirm the device's excellent beam directivity and precise beam-steering capability, both essential for OWC systems. To assess operational stability in OWC scenarios, tests are conducted at a fixed distance of 1 m. Figure 3f shows the stability measurement results, demonstrating a gradual decrease in received optical power over time, with 3 dB attenuation occurring after 174 min. This confirms that the OPA maintains high stability, which is crucial for reliable OWC applications.

## High-speed OWC system based on OPA

To validate the high-speed communication performance of the OPA, a coherent detection-based measurement system is employed. This system forms the basis of the experimental setup illustrated in Fig. 4a, which is designed to evaluate two key functionalities of the OPA: optical beam steering and high-speed signal transmission. In this configuration, the OPA directs optical beams toward six distinct angles ($\theta = 0°, \varphi = 1.6°; \theta = 0°, \varphi = 3.4°; \theta = 0°, \varphi = 4.8°; \theta = 31°, \varphi = 1.6°; \theta = 31°, \varphi = 3.4°; \theta = 31°, \varphi = 4.8°$) while transmitting 320-Gbps 16-QAM signals for analysis. Since the OPA's far-field divergence is symmetric in the transverse direction, performance is characterized over half of the full angular span. The longitudinal range is limited to 3.2° due to the experimental setup optimized for the C-band in high-speed coherent transmission, which represents an instrumentation constraint and not a fundamental limitation of the OPA. By integrating beam steering with high-speed data transmission tests, this approach provides a comprehensive assessment of OPA performance under realistic communication conditions. Such evaluation is essential to verify whether the OPA can maintain reliable high-speed communication during beam steering, a fundamental requirement for practical OWC systems. The detailed experimental procedure is as follows. A laser source operating at 1550 nm emits a continuous wave optical signal, which is amplified using an erbium-doped fiber amplifier (EDFA). A PC adjusts the polarization state before an optical splitter divides the signal into two paths: one for electro-optic modulation and the other serving as the local oscillator (LO) at the receiver. A digital-to-analog converter (DAC) operating at 100 GSa/s generates a 320-Gbps 16-QAM electrical signal. These I/Q parts are amplified by two electrical amplifiers (EAs) before driving a 35 GHz in-phase/quadrature modulator (IQM). The IQM is biased at the transmission null to suppress the carrier component, thereby enabling efficient modulation of the optical signal in the signal arm[41]. The modulated light is further amplified by another EDFA and then injected into the OPA for beam steering at predefined angles.

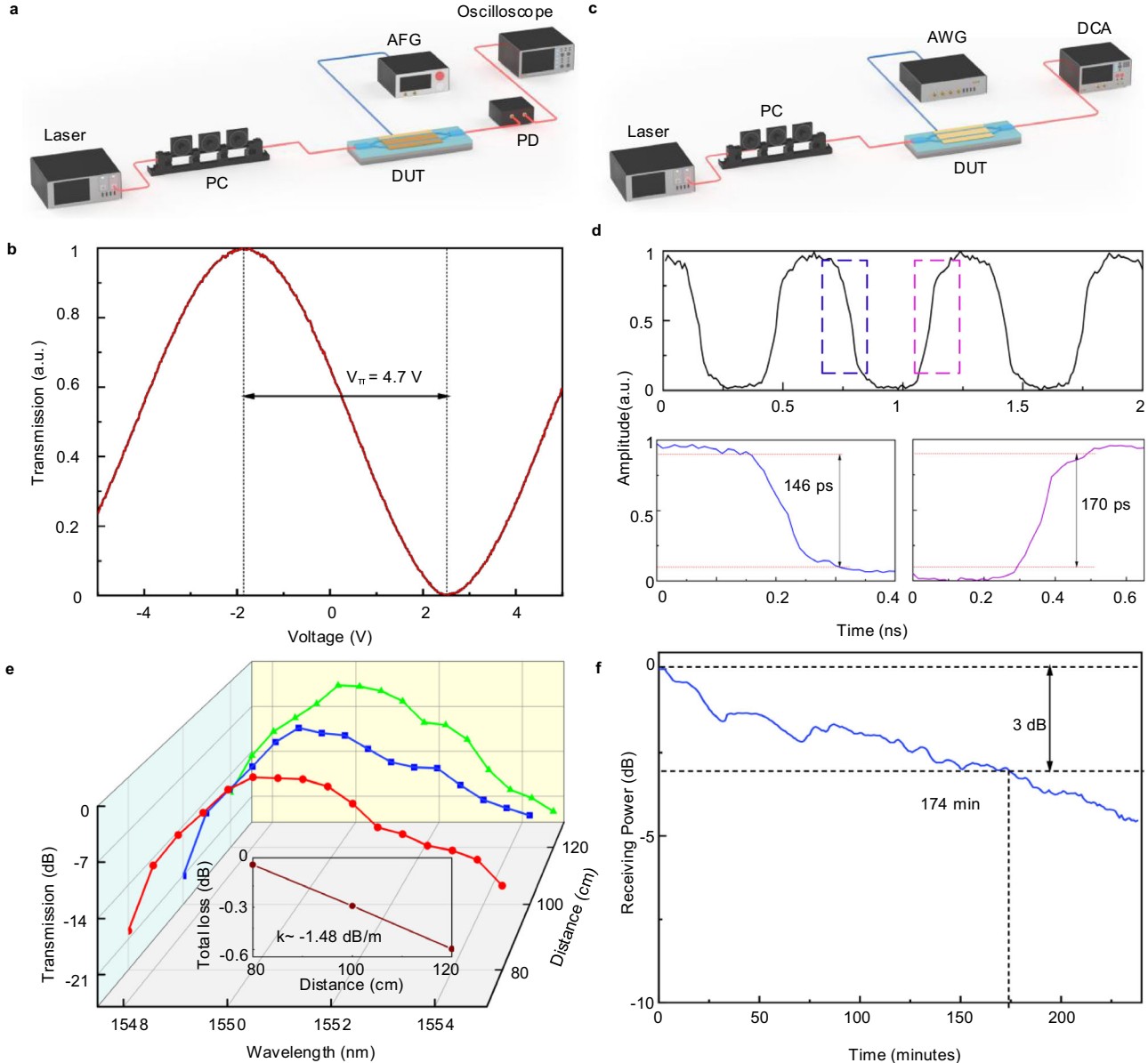

**Fig. 3 | Performance characterization of the OPA. a** Experimental setup for measuring the half-wave voltage ($V_\pi$) of the OPA phase shifter. **b** Normalized optical transmission as a function of driving voltage. **c** Experimental setup for measuring the rise and fall times of the OPA phase shifter. **d** Measured time-domain response of the modulated optical signal under a 1 GHz square-wave driving voltage. **e** Transmission efficiency of the OPA-based link at different spatial transmission distances. The inset presents the corresponding linear fitting result. **f** Temporal variation of received optical power during the OPA operational stability test. PC polarization controller, DUT device under test, AFG arbitrary function generator, PD photodetector, AWG arbitrary waveform generator, DCA digital communication analyzer, $V_\pi$ half-wave voltage, $k$ attenuation rate.

After 93 cm of free-space propagation, the beam is collected by a collimator and coupled back into a fiber. At the receiver, the EDFA is employed for compensating the transmission loss. The received carrier-suppressed modulated signal is detected with a single-polarization coherent receiver. To meet the polarization-matching requirement in a coherent receiver, two PCs ensure precise alignment between the signal and the LO relative to the optical hybrid. Finally, the output photocurrents are captured using a digital storage oscilloscope (DSO) with a sampling rate of 160 GSa/s.

To illustrate the operational principles of signal generation and recovery in our high-capacity transmission system, Fig. 4b depicts the digital signal processing (DSP) flowchart for the 320 Gbps 16-QAM system. At the transmitter, the data stream is upsampled and pulse-shaped using a root-raised cosine (RRC) filter with a roll-off factor of

0.01. These steps precede DAC and optical modulation. On the receiver side, coherent detection is first applied. The captured signal is then resampled and subjected to a series of digital compensation steps, including frequency offset correction, matched filtering, and equalization based on a multiple-input multiple-output feed-forward equalizer (MIMO-FFE). After the linear equalization-enhanced noise following these compensations, carrier phase recovery is conducted. Additional noise suppression is achieved through a post-filter (PF) combined with maximum likelihood sequence detection (MLSD), enhancing BER performance. This comprehensive DSP workflow ensures high-quality generation and accurate recovery of the 320 Gbps 16-QAM signals, which is essential for sustaining the reliability of our high-capacity transmission system. Further details on the complete DSP procedure are provided in Supplementary Note 9.

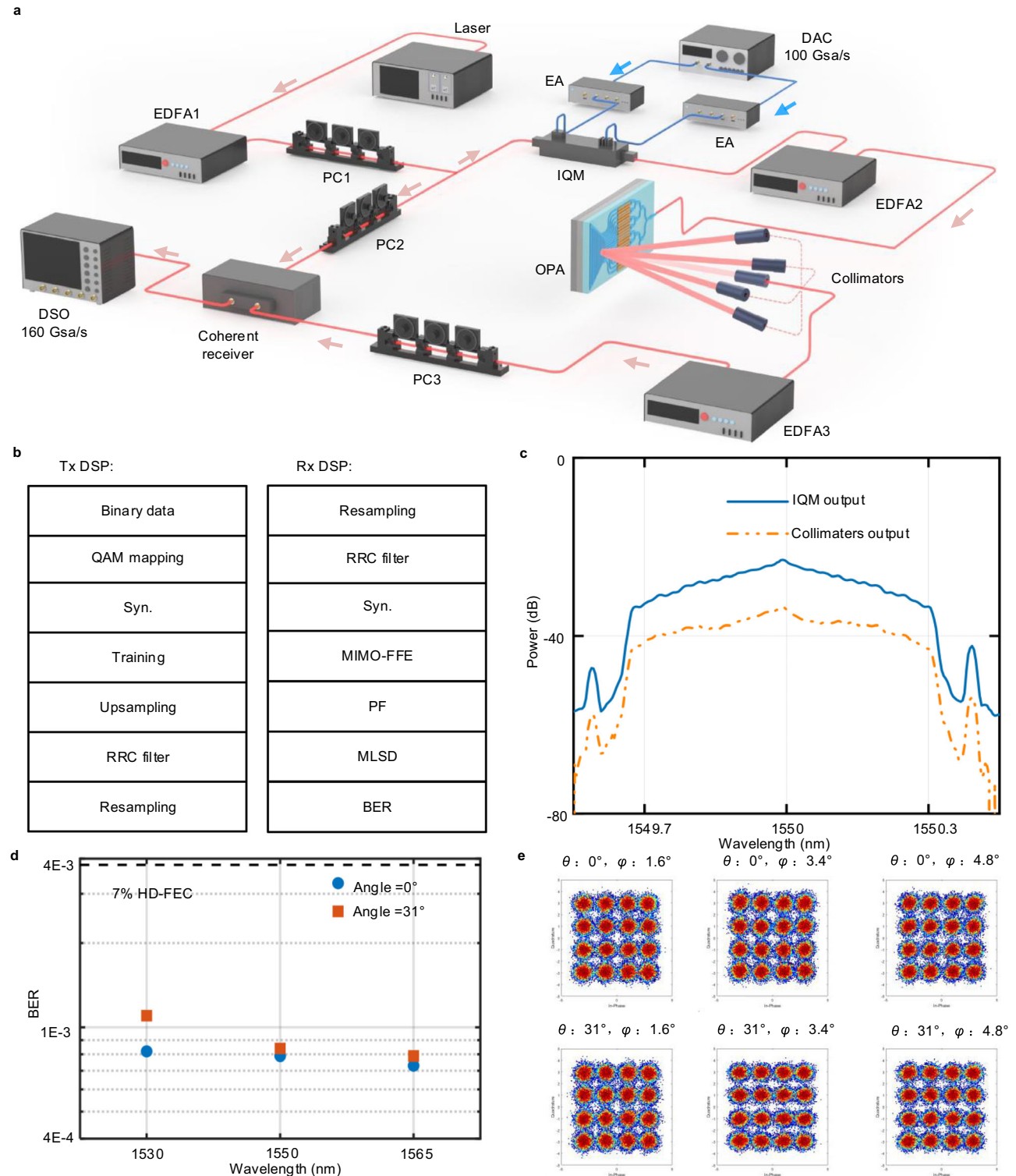

**Fig. 4 | Experimental results of a high-speed OWC system utilizing the OPA.**
**a** Schematic of the 320 Gbps data transmission link. **b** DSP flowchart implemented at the transmitter and receiver. **c** Optical spectra of the 16-QAM signal recorded at different points within the system. **d** BERs measured at different wavelengths for transverse angles $\theta = 0°$ and $\theta = 31°$, respectively. **e** Recovered constellation diagrams corresponding to each angle. PC polarization controller, EDFA erbium-doped fiber amplifier, IQM in-phase/quadrature modulator, EA electrical amplifier, DAC digital-to-analog converter, OPA optical phased array, DSO digital storage oscilloscope, QAM Quadrature Amplitude Modulation, RRC root-raised cosine, Syn. Synchronization, MIMO-FFE multiple-input multiple-output feed-forward equalizer, PF post-filter, MLSD maximum likelihood sequence detection, BER bit error rate, HD-FEC hard-decision forward error correction.

To characterize the spectral evolution of the transmitted signal and evaluate link-induced impairments, Fig. 4c presents high-resolution spectral measurements (1.12 pm) of the 320 Gbps 16-QAM signal acquired at different stages of the transmission link. The fine spectral resolution enables clear visualization of changes introduced during modulation, amplification, and transmission. These measurements provide direct evidence for assessing signal integrity and link-induced losses. To evaluate system stability under multi-angle beam

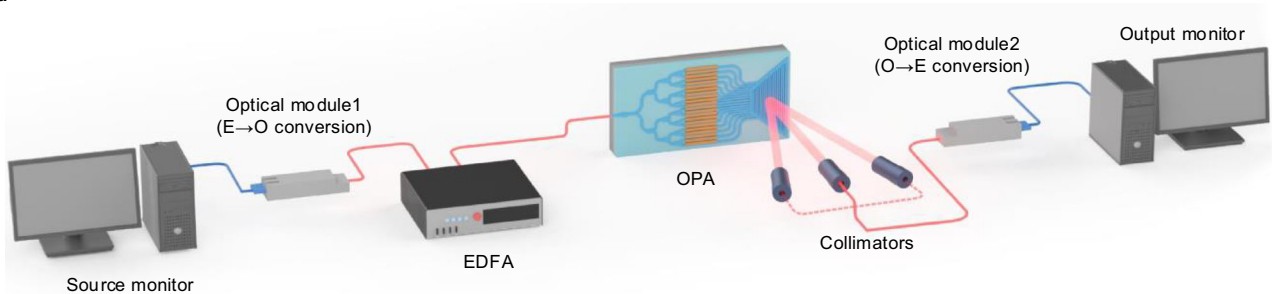

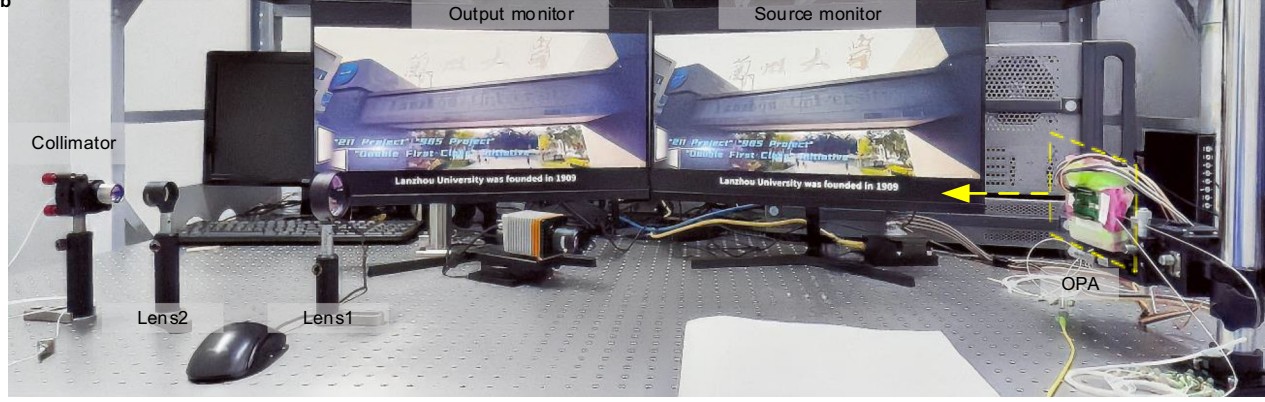

**Fig. 5 | Real-time HD video transmission in an OWC system. a** Schematic of the point-to-point experimental link. **b** Photograph of the experimental setup. Real-time HD video transmission is demonstrated when the receiver is oriented at 0°, with the yellow arrow indicating the direction of optical signal propagation. EDFA erbium-doped fiber amplifier, OPA optical phased array.

steering via the OPA, Fig. 4d shows the BER measured at different wavelengths for transverse angles $\theta = 0°$ and $\theta = 31°$, respectively. The BER remains consistently below the 7% hard-decision forward error correction (HD-FEC) threshold of $3.8 \times 10^{-3}$ for every angle, with a maximum observed value of only $1.1 \times 10^{-3}$. This confirms robust system stability during OPA operation at various steering angles, indicating that beam steering does not introduce significant signal degradation and attests to the system's reliability in dynamic beam control scenarios. To further assess signal quality across steering angles, Fig. 4e displays recovered 16-QAM constellation diagrams obtained at different angles. These constellations consistently demonstrate high signal integrity across all channels. Further enhancements in transmission capacity may be pursued through several strategies. Increasing the performance data rate of the signal source can alleviate the DAC and DSO current rate limitation, thereby boosting overall transmission capacity. Incorporating an optical bandpass filter (OBPF) after the second EDFA could suppress amplified spontaneous emission (ASE) noise, improving signal quality and facilitating higher data rates or extended reach. Furthermore, lithium niobate exhibits marked optical anisotropy, enabling independent propagation and manipulation of distinct polarization states. Exploiting defined polarization states such as transverse magnetic (TM) mode significantly expands the scanning range of OPA, thereby increasing the channel capacity in OWC systems (see Supplementary Note 10).

### Wireless video transmission based on OPA

After characterizing the optical properties and validating the communication performance of the core device, we construct an OWC system based on the OPA. This system is employed to perform point-to-point HD video transmission experiments. To illustrate the system design and experimental configuration, Fig. 5a, b present the schematic diagram and the actual experimental setup, respectively. The specific experimental procedure is as follows. A computer controls an optical module (electrical-to-optical conversion) to convert the electrical signal from an HD video stream on the source monitor into an optical signal. This signal is coupled into a single-mode fiber for transmission. The optical signal is then directed through an EDFA for power amplification before being coupled into the OPA chip. Under active control, the OPA emits the optical signal directionally at a steering angle of 0°. After 93 cm of free-space propagation, the optical signal is collected by a collimator at the receiver. The collected signal is delivered to another optical transceiver module (optical-to-electrical conversion), which reconstructs the original analog electrical signal. This process ultimately results in stable, undistorted HD video output on the output monitor, confirming that the OPA-based OWC system supports high-quality signal transmission.

To evaluate the reliability of the OWC system, we conduct tests of HD video transmission. Experimental results demonstrate that the HD video stream remains stable and uninterrupted when the beam pointing angle of the transmitter aligns with that of the receiver (see Supplementary Movie 1). This finding confirms that precise beam alignment between the transmitter and receiver is essential for maintaining a stable communication link. Conversely, the communication link is immediately interrupted when the free-space channel between the OPA transmitter and the receiver is obstructed (see Supplementary Movie 2). Notably, the OPA device employed in this study enables dynamic beam steering to any specified angle through wavelength or phase control. This capability facilitates direct parallel signal transmission in multiple directions without the need for additional multiplexing or demultiplexing components, thereby simplifying the system architecture and reducing hardware complexity. It also allows flexible coverage of any designated communication area, enhancing the system's adaptability for practical OWC applications such as multi-user communication or dynamic coverage adjustment.

### Discussion

This study presents a multi-target and ultra-high-speed OWC system utilizing a TFLN-based OPA. The system demonstrates exceptional

dynamic beam-steering performance, achieved through an optimized passive antenna architecture and the inherent advantages of TFLN as an active material. At the passive level, a curved coupled trapezoidal grating antenna array mitigates key limitations of conventional OPAs. Traditional designs often suffer from phase discontinuities caused by grating ridges along the phase modulation dimension, which constrain the FOV and degrade the SLSR. The proposed curved coupling structure minimizes waveguide spacing while maintaining low inter-element crosstalk. Furthermore, the trapezoidal antenna geometry effectively suppresses edge scattering. Together, these innovations enable a wide FOV of 62° × 11° and a low SLSR of −13.6 dB. This FOV is comparable to that of large-scale mechanical steering systems and significantly exceeds that of most chip-scale TFLN OPAs. The achieved SLSR also surpasses values reported for OWC systems based on $Si_3N_4$–Si OPA (−6 dB[42]) and silicon-based OPA (−7.2 dB[43]). This structural design ensures high-precision beam steering, which is essential for multi-target tracking and interference suppression in dense user environments. By independently controlling the transverse and longitudinal dimensions, a 4 × 4 point-cloud matrix is generated within a 40° × 10° FOV, successfully reconstructing the patterns of the letters "L", "Z", and "U". Experimental results confirm high directional accuracy and beam stability, supporting the system's applicability in optical wireless links. At the active level, the system leverages the strong Pockels effect in TFLN to achieve picosecond-order beam steering with exceptionally low power consumption. The phase shifters exhibit measured rise and fall times of 170 ps and 146 ps, respectively, yielding an average transition time of 158 ps, and a modulation efficiency of 8.73 pJ/π. This steering speed is orders of magnitude faster than that of OWC systems based on silicon-based OPAs (24.4 µs[9]) and MEMS-based systems, which typically operate on millisecond timescales[44]. It therefore enables real-time beam switching among multiple targets without latency. The system also consumes significantly less power, dramatically reducing the power consumption compared to the 20 mW/π reported for silicon-based OPA OWC systems[9] and the 1.33 W/π of $Si_3N_4$–Si OPA OWC systems[42].

The TFLN OPA-based OWC system establishes a record-breaking communication performance, as demonstrated via high-speed data transmission and HD video streaming experiments. Employing 16-QAM modulation, the system attains a single-channel data rate of 320 Gbps across six distinct steering angles within the FOV of 31° × 3.2°. This rate significantly surpasses the peak capabilities of 5 G and existing 6 G proposals, representing the highest single-channel rate reported to date for OWC systems. For context, recent metasurface-based OWC systems achieve up to 200 Gbps[11], whereas silicon-based OPAs are limited to 40 Gbps[43]. The BER across all angles remains below $1.1 \times 10^{-3}$, well under the 7% HD-FEC threshold of $3.8 \times 10^{-3}$. This consistency confirms that beam steering does not introduce significant signal distortion, a critical requirement for dynamic multi-user communication. The system further demonstrates practical applicability through stable transmission of uncompressed HD video. In contrast to earlier OWC video transmission systems that rely on mechanical components[45] or static metasurfaces[11], our design enables real-time video delivery without moving parts, thereby reducing complexity and enhancing reliability. Video transmission continues uninterrupted when the transmitter and receiver are aligned and resumes instantaneously upon restoration of the free-space channel. These attributes are essential for applications such as indoor LiFi and drone communications, where robust and immediate link recovery is imperative. As summarized in Table 1, the proposed system outperforms existing OWC technologies across several key metrics. Relative to MEMS-based systems[44], it exhibits higher steering speed, superior data rates, and full on-chip integrability. In comparison to metasurface-based systems[11,12,46], it supports dynamic beam scanning, improved SLSR, and higher modulation efficiency. When evaluated against other OPA-based OWC systems[9,42,43], our design achieves

**Table 1 | Comparison of representative works in high-speed OWC**

| Year | Method | On-chip | Dynamic beam scanning | SLSR | FOV | Single-channel speed | Switching speed | Modulation efficiency | Applications |
|---|---|---|---|---|---|---|---|---|---|
| 2016 (ref. 44) | MEMS | Yes | Yes | NA | 4° | 10 Gbps (16 QAM) | NA | NA | Signal |
| 2018 (ref. 45) | AWGR | No | No | NA | 18.6° × 18.6° | 35 Gbps (OOK) 112 Gbps (PAM 4) | NA | NA | Signal/Video |
| 2020 (ref. 61) | LCoS-SLM | No | Yes | NA | 6° | 60 Gbps (PAM 4) | NA | NA | Signal |
| 2020 (ref. 62) | Metasurface & AWGR | No | No | NA | 35° | 20 Gbps (OOK) | NA | NA | Signal |
| 2022 (ref. 12) | Metasurface | No | No | NA | 80° | 100 Gbps (OOK) | NA | NA | Signal |
| 2023 (ref. 46) | Metasurface | No | No | NA | 20° × 20° | 100 Gbps (QPSK) | NA | NA | Signal |
| 2023 (ref. 63) | Metasurface & LCoS-SLM | No | Yes | NA | 20° × 20° | 10 Gbps (OOK) | ~ms | NA | Signal |
| 2023 (ref. 64) | Metalens & Translation stage | No | No | NA | 80° | 10 Gbps (OOK) | NA | NA | Signal |
| 2024 (ref. 11) | Metasurface | No | No | NA | 120° | 200 Gbps (OOK) | NA | NA | Signal/Video |
| 2024 (ref. 21) | Topological beamformer | Yes | No | NA | 360° | 72 Gbps (16 QAM) | NA | NA | Signal/Video |
| 2024 (ref. 9) | Silicon-based OPA | Yes | Yes | NA | 60° × 14° | 12 Gbps (OOK) | 24.4 µs | 20 mW/π | Signal/Image |
| 2024 (ref. 42) | $Si_3N_4$- Si-based OPA | Yes | Yes | −6 dB | 96° × 14.4° | NA | 32.26 µs | 1.33 W/π | Image |
| 2025 (ref. 43) | Silicon-based OPA | Yes | Yes | −7.2 dB | 25.6° × 8.1° | 40 Gbps (OOK) | NA | 1.92 W (total) | Signal |
| This work | TFLN-based OPA | Yes | Yes | −13.6 dB | 62° × 11° | 320 Gbps (16 QAM) | 158 ps | 8.73 pJ/π | Signal/Video |

NA not measured, MEMS micro-electrical mechanical system, AWGR arrayed waveguide grating router, LCoS-SLM liquid crystal on silicon spatial light modulator, OPA optical phased array, OOK on-off-keying, PAM 4 pulse amplitude modulation 4-level, 16QAM 16-Quadrature Amplitude Modulation, QPSK quadrature phase-shift keying.

a wider FOV, lower sidelobe levels, faster switching speed, and significantly reduced power consumption. A detailed comparison with other relevant TFLN OPAs is provided in Supplementary Table 1.

The proposed TFLN-based OPA creates new opportunities for high-performance OWC across multiple domains. In indoor environments, it serves as a high-speed LiFi access point, providing multi-gigabit connectivity to numerous users concurrently while avoiding radio frequency interference. For unmanned aerial vehicles (UAVs), the system combines low power consumption with rapid beam steering, supporting secure, high-bandwidth data links between drones and ground stations—features particularly beneficial for applications such as aerial monitoring and disaster management. In satellite communications, the fully solid-state construction and wide FOV enable the development of compact and lightweight laser communication terminals suitable for inter-satellite linking, a critical requirement for next-generation satellite constellations. Beyond communications, the ability to generate programmable beam patterns suggests potential in optical imaging and LiDAR systems. The integration of OWC and LiDAR functionalities on a single TFLN platform may pave the way for multifunctional devices in autonomous vehicles or robotics, where simultaneous high-speed data transmission and precise environmental sensing are imperative.

Achieving complete miniaturization and autonomous operation of terminal devices represents a critical objective for next-generation OWC systems. A promising approach is the development of highly integrated on-chip light sources. Although existing systems utilizing external laser sources have successfully demonstrated architectural feasibility[47,48], future advances are likely to employ hybrid integration of high-performance quantum dot lasers or heterogeneous integration techniques to realize compact, low-power integrated photonic terminals. Such progress significantly enhances practical utility and compatibility, aligning with evolving trends in solid-state LiDAR technology. The TFLN platform offers exceptional potential for heterogeneous integration, providing a robust and versatile foundation for addressing on-chip integration challenges. Significant performance enhancements can be realized through several advanced technical pathways. For instance, the heterogeneous integration of optical frequency combs enables multi-wavelength parallel transmission, dramatically increasing channel capacity and spectral efficiency[49]. Meanwhile, on-chip integrated erbium-doped waveguide amplifiers (EDWAs) can effectively compensate for link loss and enhance output optical power[50,51]. In practical deployment of OPAs, the primary factor limiting transmission distance is the beam divergence at the optical power output. Therefore, optimizing antenna design and output coupling schemes, such as apodized gratings[52,53], metasurface pre-collimators[54], integrated microlens arrays[55], and dynamic beam shaping control[25,56], will achieve tighter beam confinement and higher far-field collimation efficiency, thereby enabling long-range, low-power optical links. More importantly, co-integrating key functional units such as photodetectors will ultimately lead to a fully functional system-on-a-chip (SoC) that incorporates transmission, modulation, and reception[57,58]. Such integration markedly reduces the size, weight, and power consumption of the system while substantially improving reliability, meeting the stringent hardware requirements of future ubiquitous access networks. While the present work demonstrates a 16-channel OPA as a proof of concept, we recognize that scaling the channel count to several hundred or more is essential for applications demanding both wide FOV and fine angular resolution. Compared with silicon photonics, lithium niobate waveguides exhibit a lower refractive index contrast, which historically has limited integration density. However, the strong Pockels effect in lithium niobate enables high-speed, low-power phase modulation without free-carrier absorption or thermal crosstalk burdens, offering a distinct pathway for scalable, energy-efficient systems. Fabrication processes such as deep ultraviolet lithography have already achieved low propagation loss in TFLN,

confirming the feasibility of maintaining phase uniformity across large arrays[59]. Furthermore, hybrid integration with low-loss passive waveguides and innovative electronic driving architectures, including row-column addressing[60], can alleviate layout and control complexity in future large-scale implementations. With continued development in co-integration techniques and custom CMOS drivers, TFLN OPAs are positioned to achieve channel counts comparable to silicon-based systems while retaining superior switching speed and power efficiency.

At the system level, extending the transmission distance is a primary optimization objective. Current research on transmission loss provides a clear reference for further advancements. In practical applications, commercially available fiber amplifiers can be employed for effective signal compensation. Ultimate performance will benefit from the high power capability inherent to the TFLN platform. Moreover, co-optimizing antenna design with intelligent control algorithms can expand beam steering ranges and enhance angular resolution, thereby significantly improving multi-target access capability and spatial multiplexing efficiency. Exploring novel mechanisms, such as phase and wavelength dual modulation, could introduce wavelength-division multiplexing into space-division multiplexing systems. Such a strategy would exponentially increase the number of communication channels and establish new paradigms for high-speed, high-capacity, long-range optical wireless networks.

## Methods

### Simulation and fabrication of the OPA chip

The near-field and far-field distributions of the OPA antenna are simulated using the finite-difference time-domain (FDTD) method. These simulations are carried out with the commercial software Lumerical FDTD Solutions. Boundary conditions are applied in three-dimensional space, specifically along the $x$, $y$, and $z$ directions, to accurately model the periodic grating antenna array structure shown in Fig. 1b.

Lithium niobate on insulator (LNOI) wafers are used as the substrate for fabricating the chip. Waveguide structures are patterned by means of deep ultraviolet (DUV) lithography and then precisely transferred into the TFLN via dry etching. The sidewalls of the waveguides exhibit an inclination angle of approximately 67°. The measured transmission loss of the fabricated waveguide is below 0.5 dB/cm.

Chip packaging is implemented through optoelectronic hybrid integration. The fabricated chips are mounted onto custom-designed metal substrates using conductive silver adhesive. Electrical connections between on-chip pads and external printed circuit boards are established with gold wire bonding. For optical interfacing, a polarization-maintaining fiber array with a small mode field diameter is selected. The polarization direction of the fiber is carefully aligned with the transverse electric (TE) mode of the chip waveguide. This alignment minimizes mode mismatch loss and preserves the polarization state. Low-loss optical coupling between the fiber array and the on-chip optical ports is achieved through active alignment. The output of the fiber array is terminated with a standard FC/APC connector to ensure low insertion loss and low back reflection. The electrical interface employs a dual-row pin header with a 2.54 mm pitch, which offers versatile and plug-compatible connectivity to external drive and control systems.

### Experimental setup

Beam characterization of the fabricated OPA is performed using a tunable laser (Santec TSL 570), a PC (Thorlabs FPC561), an optical screen, and an infrared camera (Xenics Bobcat 640). For subsequent measurements, a triangular wave signal generated by an AFG (Tektronix AFG3102C) is applied to determine the $V_\pi$ of the phase shifter. The output signal is detected by a PD (Finisar XPDV3120R, 70 GHz) and displayed on an oscilloscope (Tektronix TDS2012C, 100 MHz). The device response time is evaluated using a square-wave signal from an

AWG (Keysight M8199B, 224 GSa/s), and the resulting waveform is analyzed with a DCA (Keysight N1000A). The OPA transmitter, driven by a voltage source controlled by a field programmable gate array (FPGA), is coupled to a fiber collimator (Thorlabs F810APC 1550). This setup is used to characterize the relationship between the received optical power, measured with a power meter (Thorlabs PM100D), and the transmission distance, as well as to assess temporal stability. Before coupling, beam steering is carried out using a plano-convex lens (Jcoptix OLC320166 T4M) and a plano-concave lens (Jcoptix OLE240136 T4M) to ensure a collimated input beam to the collimator.

In the high-speed coherent communication experiment, the output from a tunable laser (Keysight 8190A) is divided into two separate paths. One path function as the optical carrier, while the other undergoes electro-optic modulation. An IQM (FTM7992HM), driven by a 100 GSa/s DAC (Micram DAC4), generates a 16-QAM signal. The DAC output is first amplified by an EA before driving the IQM. A collimator (Thorlabs F810APC 1550) collects the free-space optical signal transmitted from the optical power amplifier. Both the modulated signal and the optical carrier are then fed into a coherent receiver. The final output is captured using a 160 GSa/s DSO (LeCroy 36Zi A).

In the point-to-point HD video transmission experiment, a computer-driven optical module (Huawei, 10 G 1550 nm 80 km SM SFP +) encodes video data into optical signals. These signals are coupled into a single fiber within a fiber array. The light emitted from the fiber facet is deflected by the OPA at programmed angles and propagates through free space over a predetermined distance. A receiver collimator (Thorlabs F810APC 1550) collects the optical signal. This signal is then routed to a second optical transceiver, where it undergoes photoelectric conversion, ultimately reconstructing the original video signal for display on an HD monitor.

## Data availability

All the data supporting the findings in this study are available in the paper and Supplementary Information. Source data are provided with this paper. Additional data related to this paper are available from the corresponding authors upon request. Source data are provided with this paper.

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

## Acknowledgements

Y.T. acknowledges the funding from the National Natural Science Foundation of China (W2411059), the Gansu Provincial Science and Technology Major Special Project (25ZDWA001, 25ZDGA005). M.Y. acknowledges the funding from the National Natural Science Foundation of China (62405125), the Key Research and Development of Gansu Province (24YFGA007), the Joint Research Fund Project of Gansu Province (25JRRA1126), and the Talent Scientific Fund of Lanzhou University.

## Author contributions

All authors contributed extensively to the work presented in this paper. X.M. conceived the study. X.M., M.Y., H.Z., and B.H. designed and characterized the OPA chip. J.L., X.M., and M.Y. conducted the optical communication experiments. P.Z., Y.J., X.M., and B.H. performed the HD video transmission experiment. X.M., M.Y., H.Z., and Y.T. conducted a theoretical analysis. X.M., M.Y., H.X., G.R., A.M., Y.S., and Y.T. wrote the manuscript with input from all coauthors. Y.T. supervised the research. All authors participated in data analysis and discussion.

## Competing interests

The authors declare no competing interests.
