## [Peer Review File · Nature Communications]

Multi-target and ultra-high-speed optical wireless communication using a thin-film lithium niobate optical phased array

Corresponding Author: Professor Yonghui Tian

Version 0:

Reviewer comments:

Reviewer #1

(Remarks to the Author)

This work presents a TFLN-based OPA for beam steering and high-speed optical wireless communications. Key results include 16-channel configuration, wide FOVs, fast steering speed, low sidelobe level, 320 Gbps OWC transmission, and HD video demonstration. The paper is very well written and the experimental works are solid. While I am very impressed by the achieved results, I do have a few major concerns regarding its novelty and significance, considering the standard of the targeted journal.

1. Compared with silicon-based OPAs, the biggest concern for TFLN-based OPAs is the compactness and scalability in channel counts. SiP based OPAs have demonstrated up to 1000 channels (e.g. <https://arxiv.org/abs/2508.19977>), thus can achieve wide FOV and fine resolution simultaneously. Also the demonstrated sidelobe level is even lower than this work within the same FOV range. Comparably, the 16-channel TFLN device presented in this work is not that impressive. The authors should critically discuss the challenges and solutions to scale up the channel numbers of TFLN-based OPAs to a few hundred or even thousand. The energy efficiency of the overall OPA also correlates with the channel numbers. Otherwise, the steering speed merit alone brought by TFLN is not enough to justify its feasibility.
2. There are three difference values of FOVs presented in the paper, i.e., $62^\circ \times 11^\circ$, $40^\circ \times 10^\circ$, $31^\circ \times 3.2^\circ$, all coming from the same device. This is a bit confusing to follow. I understand that the performance of the pattern reconstruction and high-speed transmission limit the operational range of FOV. The authors are suggested to make it clear in the beginning regarding the reasons of such limitations, and discuss them in details in corresponding sections.
3. Using curved waveguides to address the different pitch mismatch between the phase shifter section and the optical antenna section is indeed a viable design. What's the maximum time delays induced by the different waveguide lengths in this design? And please also discuss how much it impacts high speed signal transmission.
4. Three EDFAs are used for the transmission experiment, yet the transmission distance is only 1m. In the discussion section, the authors repeatedly suggest that extending the distance should rely on more amplifications, either EDWAs or fiber amplifiers. In my opinion this is not exactly where the issue is and shouldn't be considered as the way forward. Considering the atmospheric loss in this wavelength should be well below 1 dB/km once the beam is properly collimated, a more precise beam divergence control at the output of the OPA should be the key issue to solve for practical deployment.
5. There is a typo in Fig. 4a, 'DCA' should be 'DAC'.

Reviewer #2

(Remarks to the Author)

This manuscript presents a multi-target, ultra-high-speed optical wireless communication system based on a thin-film lithium niobate optical phased array. The system demonstrates impressive performance, including high-speed multi-target beam steering across a wide field of view, low sidelobe levels, and a record single-channel data rate of 320 Gbps using 16-QAM modulation. The work highlights significant advantages in integration, power efficiency, and dynamic reconfigurability,

supported by comprehensive experimental data such as high-definition video transmission. The results are clearly presented and address a highly relevant research direction with considerable potential for next-generation wireless networks. The manuscript is well-structured, exhibits strong originality, and the data are systematically organized, providing robust support for the conclusions. This work is likely to attract broad interest from the photonics and communications communities. In my view, the paper merits publication after the authors address the following points:

1. The current vertical beam steering range is 11° . The authors should discuss potential strategies or design modifications to further expand the vertical field of view in future implementations.
2. The manuscript does not explicitly clarify the parameters governing the horizontal and vertical beam resolution, nor the ultimate factors limiting these resolutions. A discussion on these aspects would help readers better understand the system's performance constraints.
3. For multi-wavelength parallel beam control scenarios, could the authors comment on whether the SPGD algorithm requires separate objective functions for each wavelength? If a unified objective function is used, how is convergence stability maintained? Additionally, is sequential calibration necessary for the dual-wavelength operation demonstrated in the experiments?
4. Prior to operation, does the SPGD algorithm require calibration at each beam steering angle? If so, could this initialization process limit the overall communication speed or target switching rate?
5. The discussion mentions a "transition time of approximately 158 ps", while the measured rise and fall times are 170 ps and 146 ps, respectively. The calculation or definition of the 158 ps value should be clarified for consistency.
6. On page 13, first paragraph, "tcoherent detection" appears to be a typographical error. This should be corrected to "coherent detection".
7. Please verify the angle units in Figure 4d and ensure consistent capitalization of units in Table 1 (e.g., 'mW' vs. 'mw').

Version 1:

Reviewer comments:

Reviewer #1

(Remarks to the Author)

I have carefully gone through the authors' response and the revisions to the manuscript. In my opinion, the revised manuscript has convincingly addressed my previous comments and concerns. I don't have more suggestions and would like to recommend accepting it for publication.

Reviewer #2

(Remarks to the Author)

The authors have addressed all my concerns and I recommend the publication of the manuscript.

made.

Response to the reviewers' comments

To Reviewer #1's comments:

Recommendation: This work presents a TFLN-based OPA for beam steering and high-speed optical wireless communications. Key results include 16-channel configuration, wide FOVs, fast steering speed, low sidelobe level, 320 Gbps OWC transmission, and HD video demonstration. The paper is very well written and the experimental works are solid. While I am very impressed by the achieved results, I do have a few major concerns regarding its novelty and significance, considering the standard of the targeted journal.

Author Response:

Thank you for your positive evaluation and constructive feedback. We are grateful for the opportunity to further clarify the novelty and significance of our work.

To our knowledge, compared with silicon, lithium niobate exhibits a strong linear electro-optic effect (Pockels effect), which enables pure phase modulation of light with **higher speed and lower consumption** by applying a traveling-wave electrode along both sides of the lithium niobate waveguide, without requiring carrier injection or thermo-optic effects. For example, **our fabricated OPA demonstrates a switching speed of 158 ps and a modulation efficiency of 8.73 pJ/ π , all of which are among the best performances reported for OPAs to date, whether lithium niobate-, silicon- or MEMS-based.** These results pave the way for high-speed, low-power optical wireless communication (OWC) systems.

In this work, we also introduce **a new integrated OPA architecture on the TFLN platform, which incorporates curved couplers and trapezoidal grating antennas to mitigate phase discontinuity.** Thanks to this novel design, the fabricated 16-channel OPA device can achieve two-dimensional beam steering with a $62^\circ \times 11^\circ$ FOV and a low sidelobe level of -13.6 dB. Notably, previously reported systems with comparable performance typically require 32 or more phase channels, underscoring the superior compactness of our device.

This work also presents the first-ever experimental demonstration of a 320 Gbps optical wireless link using 16-state quadrature amplitude modulation together with OPA-based beam steering. This data rate substantially exceeds the throughputs of current 5G systems and the projected capabilities of early 6G proposals. System robustness is validated by real-time transmission of uncompressed high-definition video. The approach enables multi-beam control without moving parts and supports real-time multi-target tracking and communication.

In summary, the combined contributions in device architecture, record-setting performance metrics, and system-level demonstration represent a substantive breakthrough in integrated lithium niobate photonics. The results open immediate opportunities in secure mobile links, LiFi, and free space optical communication, and they create clear avenues for future applications in advanced LiDAR and computational imaging. We believe these clarifications address the reviewer's comments and convey the high novelty and academic value of the work. Thank you again for the careful review and helpful suggestions.

Comment 1:

Compared with silicon-based OPAs, the biggest concern for TFLN-based OPAs is the compactness and scalability in channel counts. SiP based OPAs have demonstrated up to 1000 channels (e.g. <https://arxiv.org/abs/2508.19977>), thus can achieve wide FOV and fine resolution simultaneously. Also the demonstrated sidelobe level is even lower than this work within the same FOV range. Comparably, the 16-channel TFLN device presented in this work is not that impressive. The authors should critically discuss the challenges and solutions to scale up the channel numbers of TFLN-based OPAs to a few hundred or even thousand. The energy efficiency of the overall OPA also correlates with the channel numbers. Otherwise, the steering speed merit alone brought by TFLN is not enough to justify its feasibility.

Reply and modifications:

We sincerely thank the reviewer for their insightful and constructive comments regarding the scalability and feasibility of our TFLN OPA. We greatly appreciate the

opportunity to address these important points and further clarify the strengths and potential of our proposed approach.

Lithium niobate waveguides exhibit weaker optical confinement than silicon waveguides, primarily due to the lower refractive index contrast between their cladding and core regions. This results in a lower integration level for lithium niobate photonics compared to silicon photonics. Nevertheless, **lithium niobate has a unique advantage in its strong linear electro-optic effect, known as the Pockels effect. This effect allows pure phase modulation with higher speed and lower power consumption when traveling-wave electrodes are used alongside the waveguides, without relying on carrier injection or thermal effects.** In our proof-of-concept demonstration, the fabricated OPA achieves a switching speed of 158 ps and a modulation efficiency of 8.73 pJ/ π phase shift. **These performance metrics are among the best reported for lithium niobate-, silicon- or MEMS-based OPAs,** which highlights the potential of our device for applications requiring extreme speed and precision, such as high-speed OWC, quantum information processing, high-resolution lidar, and real-time holographic displays. The comparative analysis in Table 1 confirms the advantages in switching speed and power consumption for high-speed OWC system, showing how this work opens new pathways for developing high-speed, low-power OWC system.

Table 1 Comparison of representative works in high-speed OWC system.

Representative works	Method	FOV	SLSR	Single-channel speed	Switching speed	Modulation efficiency	Applications
C1	Silicon-based OPA	60°×14°	NA	12 Gbps (OOK)	24.4 μ s	20 mW/ π	Signal/Image
C2	Si ₃ N ₄ - Si-based OPA	96°×14.4°	-6 dB	NA	32.26 μ s	1.33 W/ π	Image
C3	Silicon-based OPA	25.6°×8.1°	-7.2 dB	40 Gbps (OOK)	NA	1.92 W (total)	Signal
This work	TFLN-based OPA	62°×11°	-13.6 dB	320 Gbps (16 QAM)	158 ps	8.73 pJ/π	Signal/Video

Note: NA indicates not measured.

Our study also introduces a **novel integrated OPA architecture on the TFLN platform**, which uses curved couplers and trapezoidal grating antennas to effectively suppress phase discontinuities. With this innovative design, the fabricated 16-channel OPA device achieves a two-dimensional beam steering range of $62^\circ \times 11^\circ$ and a low sidelobe level of -13.6 dB. Table 2 provides a comprehensive performance comparison with other representative TFLN OPAs reported to date. This comparison clearly demonstrates significant advancements in key parameters, including FOV, beam width, sidelobe suppression ratio, power consumption, and scanning speed. **It is noteworthy that achieving similar performance in existing reports usually requires 32 or more phase channels, indicating that our device maintains excellent performance while achieving higher integration density.**

Table 2 TFLN OPA system performance comparison.

Representative works	Platform	Channels	FOV	Beamwidth	SLSR	Power Consumption	Speed
Ref. C4	TFLN	16	$50^\circ \times 8.6^\circ$	$0.73^\circ \times 2.8^\circ$	5.1 dB	330 pJ/ π	0.4 ns
Ref. C5	TFLN	16	$24^\circ \times 8^\circ$	$2^\circ \times 0.6^\circ$	10 dB	13.5 pJ/ π	0.23 ns
Ref. C6	TFLN	32	$62.2^\circ \times 8.8^\circ$	$2.4^\circ \times 1.2^\circ$	7.4 dB	1.11 nJ/ π	14.4 ns
Ref. C7	TFLN	8	50°	8°	10 dB	NA	NA
Ref. C8	TFLN-SiN	16	$22^\circ \times 5^\circ$	$1.2^\circ \times 0.2^\circ$	NA	3.2 pJ/ π	0.83 ns
Ref. C9	TFLN-SiN	16	$62^\circ \times 7.6^\circ$	$3.2^\circ \times 1.4^\circ$	20.3 dB	41.6 pJ/ π	26 ns
This Work	TFLN	16	$62^\circ \times 11^\circ$	$3.8^\circ \times 1.72^\circ$	13.6 dB	8.73 pJ/π	158 ps

We fully agree that scaling the channel count to hundreds or thousands is essential for future high-performance systems to achieve superior angular resolution and larger effective apertures, especially in applications requiring wide FOV and high resolution. **Our proposed TFLN OPA is also well suited for such scaling.** In terms of fabrication and material properties, recent progress in deep ultraviolet lithography for TFLN has achieved excellent waveguide uniformity with propagation losses below 0.042 dB/cm

[C10]. This level of precision is critical for maintaining phase consistency in large-scale arrays. Moreover, the same deep ultraviolet equipment used for high-channel-count silicon photonic devices can be directly applied to lithium niobate on insulator wafers, meaning there are no fundamental physical barriers to fabricating hundreds to thousands of channels. Regarding electronic control, both silicon-based and TFLN OPAs face common challenges in control circuit complexity and cost when scaling to thousands of channels. Reducing the number of electronic components is important to simplify design and lower system costs. Future systems will need to adopt innovative electronic driving architectures, such as row-column addressing combined with pulse-width modulation, to address these limitations [C11]. In terms of energy efficiency, the total power consumption of an OPA includes contributions from electro-optic phase shifters, thermal tuning, and drive circuits. Our TFLN electro-optic phase shifters achieve high efficiency at $8.73 \text{ pJ}/\pi$, with energy consumption primarily occurring during switching events. This differs from silicon thermal phase shifters, which require continuous power. When scaling to thousands of channels, the total modulation power increases linearly with the number of channels, but the per-channel picojoule-level power consumption of TFLN remains significantly lower than the typical milliwatt-level consumption of silicon-based OPAs. System-level energy efficiency will still need to address drive circuit power and thermal management. Silicon-based OPAs benefit from CMOS driver integration but suffer from high static power consumption due to their modulation mechanisms, whereas thermal tuning in TFLN is mainly used for operational stability and requires very low power.

We are currently exploring multiple technical pathways to enable high-channel-count scaling for TFLN OPAs. Hybrid integration techniques, which combine the high-speed phase shifting of TFLN with the low-loss power distribution of silicon or silicon nitride, offer a viable solution to increase channel density without sacrificing modulation performance [C12, C13]. To manage layout complexity, advanced architectures under development, such as multilayer interconnects and folded routing, can optimize the electro-optic footprint [C11, C14]. Additionally, monolithic or co-integration strategies that directly integrate CMOS drivers with TFLN modulators are

crucial for achieving ultimate system-level energy efficiency and scalability [C15]. Co-design of custom CMOS drivers for TFLN modulators is also progressing to address challenges in high-voltage, high-speed driving for large arrays.

In summary, while this study focuses on demonstrating the high-speed and system-level capabilities of TFLN OPAs, the material properties, maturity of fabrication processes, and compatibility with advanced integration schemes together outline a clear and feasible path toward scaling to large-scale, high-efficiency OPAs with hundreds to thousands of channels for next-generation communication and sensing applications.

In the revised manuscript, we have added the following sentences:

“While the present work demonstrates a 16-channel OPA as a proof of concept, we recognize that scaling the channel count to several hundred or more is essential for applications demanding both wide FOV and fine angular resolution. Compared with silicon photonics, lithium niobate waveguides exhibit a lower refractive index contrast, which historically has limited integration density. However, the strong Pockels effect in lithium niobate enables high-speed, low-power phase modulation without free-carrier absorption or thermal crosstalk burdens, offering a distinct pathway for scalable, energy-efficient systems. Fabrication processes such as deep ultraviolet lithography have already achieved low propagation loss in TFLN, confirming the feasibility of maintaining phase uniformity across large arrays^[65]. Furthermore, hybrid integration with low-loss passive waveguides and innovative electronic driving architectures, including row-column addressing^[66], can alleviate layout and control complexity in future large-scale implementations. With continued development in co-integration techniques and custom CMOS drivers, TFLN OPAs are positioned to achieve channel counts comparable to silicon-based systems while retaining superior switching speed and power efficiency.” (Paragraph 4, section 3)

In the revised manuscript, we have added the following references:

“65. Liu, J. et al. High-precision propagation-loss measurement of single-mode optical waveguides on lithium niobate on insulator. *Micromachines* **10**, 612 (2019).

66. 1000-channel integrated optical phased array with 180° field of view, high resolution and high scalability. (<https://doi.org/10.48550/arXiv.2508.19977>)”

(References)

References:

- [C1] Yang, S. et al. Fast-beam-switching optical phased array for moving objects in wireless optical communication networks. *Opt. Lett.* **49**, 1961-1964 (2024).
- [C2] Li, Y. et al. Integrated communication and sensing system based on Si-SiN dual-layer optical phased array. *Opt. Express* **32**, 33222-33231 (2024).
- [C3] Jiao, W. et al. Two-dimensional multi-beam steering for parallel free-space optical communication based on a silicon nitride optical phase array. *APL Photon.* **10**, 056110 (2025).
- [C4] Li, W. et al. High-speed 2D beam steering based on a thin-film lithium niobate optical phased array with a large field of view. *Photonics Res.* **11**, 1912-1918 (2023).
- [C5] Yue, G. Li, Y. Integrated lithium niobate optical phased array for two-dimensional beam steering. *Opt. Lett.* **48**, 3633-3636 (2023).
- [C6] Wang, Z. et al. Fast-speed and low-power-consumption optical phased array based on lithium niobate waveguides. *Nanophotonics* **13**, 2429-2436 (2024).
- [C7] Li, Y. et al. Design of optical phased array with low-sidelobe beam steering in thin film lithium niobate. *Optics & Laser Technology* **171**, 110432 (2024).
- [C8] Lee, W., Kwon, Y., Kim D., Sunwoo, Y., Lee, S. Hybrid integrated thin-film lithium niobate-silicon nitride electro-optical phased array incorporating silicon nitride grating antenna for two-dimensional beam steering. *Opt. Express* **32**, 9171-9183 (2024).
- [C9] Liu, J. et al. Taylor amplitude distribution for low sidelobe in thin-film lithium niobate optical phased array. *J. Lightwave Technol.* **43**, 8286-8292 (2025).
- [C10] Liu, J. et al. High-precision propagation-loss measurement of single-mode optical waveguides on lithium niobate on insulator. *Micromachines* **10**, 612 (2019).

- [C11] 1000-channel integrated optical phased array with 180° field of view, High Resolution and High Scalability. (<https://doi.org/10.48550/arXiv.2508.19977>)
- [C12] Lee, W., Kwon, Y., Kim D., Sunwoo, Y., Lee, S. Hybrid integrated thin-film lithium niobate-silicon nitride electro-optical phased array incorporating silicon nitride grating antenna for two-dimensional beam steering. *Opt. Express* **32**, 9171-9183 (2024).
- [C13] Liu, J. et al. Taylor amplitude distribution for low sidelobe in thin-film lithium niobate optical phased array. *J. Lightwave Technol.* **43**, 8286-8292 (2025).
- [C14] Liu, M., Wu, J., Fu, Z., Zheng, S., Hou, Y. Low-cost and easy-to-embed 7-bit 360° phase shifter with transitions for large-scale phased arrays. *Microw. Opt. Technol. Lett.* **66**, 34243 (2024).
- [C15] Tan, Y. et al. Micro-transfer printed thin film lithium niobate (TFLN)-on-silicon ring modulator. *ACS Photonics* **11**, 1920–1927 (2024).

Comment 2:

There are three difference values of FOVs presented in the paper, i.e., $62^\circ \times 11^\circ$, $40^\circ \times 10^\circ$, $31^\circ \times 3.2^\circ$, all coming from the same device. This is a bit confusing to follow. I understand that the performance of the pattern reconstruction and high-speed transmission limit the operational range of FOV. The authors are suggested to make it clear in the beginning regarding the reasons of such limitations, and discuss them in details in corresponding sections.

Reply and modifications:

We sincerely thank the reviewer for this insightful comment and for highlighting the potential confusion regarding the different FOV values presented in our manuscript. We appreciate the opportunity to clarify this important aspect and have carefully revised the manuscript to address these points in a more comprehensive and accessible manner.

The three FOV values ($62^\circ \times 11^\circ$, $40^\circ \times 10^\circ$, and $31^\circ \times 3.2^\circ$) indeed originate from the same OPA device, but they correspond to distinct operational scenarios and experimental demonstrations. We acknowledge that the initial presentation may be

unclear, and we now provide explicit explanations from the beginning of the manuscript to elucidate the reasons for these variations.

Specifically, the differences arise from the following considerations:

1. The full device-level FOV of $62^\circ \times 11^\circ$ represents the maximum beam steering capability of our OPA, as determined through extensive far-field scanning under combined phase and wavelength control, as presented in Figure 2. This value defines the fundamental steering range governed by the antenna pitch and the wavelength tuning range.

2. The pattern reconstruction FOV of $40^\circ \times 10^\circ$ is selected for the point cloud reconstruction experiment because this specific angular area is necessary to project a recognizable and coherent 4×4 pattern, such as the "LZU" image. The choice is driven by the spatial requirements of the demonstration itself.

3. The high-speed communication FOV of $31^\circ \times 3.2^\circ$ is constrained by experimental setup limitations. Due to the symmetric far-field divergence distribution of the OPA in the transverse direction (as shown in Figure C1), we adopt a methodology consistent with previous research [C16], evaluating performance only within half of the full angular span during these tests. The longitudinal extent is limited to 3.2° because our high-speed coherent transmission experiments are conducted within the C-band, where our external amplifier chain and coherent receiver are specifically optimized. This limitation stems from instrumentation constraints and is not an inherent restriction of the OPA device.

Figure C1. Far-field beam spots measured at transverse steering angles of $\pm 31^\circ$, $\pm 24^\circ$, $\pm 18^\circ$, $\pm 12^\circ$, $\pm 6^\circ$, and 0° at a wavelength of 1550 nm.

In the revised manuscript, we have added the following sentences:

“Notably, we report two distinct FOV values for the same OPA device. The maximum device-level FOV of $62^\circ \times 11^\circ$ is characterized from far-field scans under

combined phase and wavelength control. The demonstration of pattern reconstruction is conducted within a $40^\circ \times 10^\circ$ FOV, which is selected as it provides the necessary angular area to project a clearly defined 4×4 point cloud pattern. (Paragraph 4, Section 2.1)

“Since the OPA’s far-field divergence is symmetric in the transverse direction, performance is characterized over half of the full angular span. The longitudinal range is limited to 3.2° due to the experimental setup optimized for the C-band in high-speed coherent transmission, which represents an instrumentation constraint and not a fundamental limitation of the OPA.” (Paragraph 1, Section 2.3)

References:

[C16]. Li, Y. et al. High-data-rate and wide-steering-range optical wireless communication via nonuniform-space optical phased array. *J. Lightwave Technol.* **41**, 4933-4940 (2023).

Comment 3:

Using curved waveguides to address the different pitch mismatch between the phase shifter section and the optical antenna section is indeed a viable design. What's the maximum time delays induced by the different waveguide lengths in this design? And please also discuss how much it impacts high speed signal transmission.

Reply and modifications:

We sincerely thank the reviewer for this insightful comment and for highlighting the importance of the curved waveguide design in our OPA. The reviewer's question regarding the maximum time delay and its impact on high-speed signal transmission is highly valuable, as it allows us to further clarify a key aspect of our device performance.

In our design, the curved waveguides address the pitch mismatch between the phase shifter section and the optical antenna section by enabling adiabatic compression of the channel pitch to approximately $1.5 \mu\text{m}$. This approach is essential for maintaining device compactness and minimizing optical crosstalk between adjacent channels. The symmetrical configuration of these curved waveguides introduces a maximum differential optical path length of about $300 \mu\text{m}$ between the outermost and innermost

channels. With a group index of 2.2 for our TFLN waveguides, this corresponds to a maximum time delay of approximately 2.2 ps.

We have thoroughly investigated the impact of this delay on high-speed signal transmission, particularly for the 320 Gbps 16-QAM signals demonstrated in our work. The symbol period for this data rate is 12.5 ps. The 2.2 ps delay represents a small portion of the symbol period (17.6 percent), and it is critical to emphasize that this delay is static and fixed for a given device configuration. This static nature leads to two considerations. First, since the delay is less than one full symbol duration, it does not cause inter-symbol interference by overlapping adjacent symbols in time. Second, as all channels operate coherently under a common optical carrier, the sub-picosecond delay differences result only in fixed phase offsets. These offsets are fully compensated during the initial beam calibration process, as described in Supplementary Note 5, and are further mitigated by standard digital signal processing algorithms implemented in our coherent communication system. Therefore, the delay does not introduce measurable signal distortion or phase misalignment in high-speed operation.

Our experimental results strongly support this analysis. The bit-error-rate performance presented in Figure 4d remains well below the hard-decision forward error correction threshold across all beam steering angles, confirming that the curved waveguide design has no observable adverse effect on communication performance.

To provide additional clarity in the manuscript, we have included the following sentences in the revised Supplementary Information:

“The curved interconnection waveguides, essential for pitch matching, introduce a maximum static path length difference of approximately 300 μm , corresponding to a time delay of about 2.2 ps. This delay is about 17.6% of a symbol duration for the 320 Gbps 16-QAM signal and therefore does not induce inter-symbol interference by overlapping adjacent symbols in time. Furthermore, the fixed phase offsets arising from these path differences are effectively calibrated out during the initial beam-forming process and are compensated by standard digital signal processing in the coherent receiver, ensuring they have no measurable impact on communication performance.”

(Paragraph 3, Supplementary Note 9, Supplementary information)

Comment 4:

Three EDFAs are used for the transmission experiment, yet the transmission distance is only 1m. In the discussion section, the authors repeatedly suggest that extending the distance should rely on more amplifications, either EDWAs or fiber amplifiers. In my opinion this is not exactly where the issue is and shouldn't be considered as the way forward. Considering the atmospheric loss in this wavelength should be well below 1 dB/km once the beam is properly collimated, a more precise beam divergence control at the output of the OPA should be the key issue to solve for practical deployment.

Reply and modifications:

We sincerely thank the reviewer for their insightful and constructive comments, which have greatly helped us improve the discussion on extending transmission distance in OWC systems based on OPAs. We fully agree with the reviewer that the key challenge for practical long-range deployment lies in achieving precise beam divergence control at the OPA output, rather than relying solely on amplification. This perspective is crucial, and we have carefully revised the manuscript to reflect this emphasis.

In our experimental setup, the three EDFAs are used primarily to compensate for insertion losses from external components, such as the in-phase/quadrature modulator, optical splitters, and fiber coupling interfaces, not to overcome free-space attenuation. As the reviewer correctly points out, atmospheric loss at 1550 nm is well below 1 dB/km with proper beam collimation. The 1 m transmission distance in our study was selected to validate high-speed modulation and coherent detection performance in a controlled laboratory setting, not to maximize link range. We acknowledge that for real-world applications, reducing beam divergence is essential.

To address this, we have expanded the discussion in the revised manuscript to clearly state that future improvements should focus on beam control over amplification. Specifically, we now highlight the following approaches as effective pathways for optimization:

1. Antenna engineering for narrower beam divergence: Refining grating geometries, including apodized or chirped gratings [C17, C18], and improving phase uniformity across channels can significantly enhance far-field beam confinement.

2. Hybrid optical packaging: Integrating on-chip microlenses or low-loss dielectric metasurfaces enables pre-collimation directly at the chip facet, reducing dependence on bulk optics [C19, C20].

3. Dynamic beam-shape control: Techniques such as adaptive phase distribution or amplitude apodization can help maintain a constant spot size over extended distances [C21, C22].

In the revised manuscript, we have modified the following sentences to emphasize beam divergence control:

“In practical implementations, this attenuation can be compensated by achieving more precise beam divergence control at the OPA output.” (Paragraph 3, Section 2.2)

“Meanwhile, on-chip integrated erbium-doped waveguide amplifiers (EDWAs) can effectively compensate for link loss and enhance output optical power [55,56]. In practical deployment of OPAs, the primary factor limiting transmission distance is the beam divergence at the optical power output. Therefore, optimizing antenna design and output coupling schemes, such as apodized gratings [57,58], metasurface pre-collimators [59], integrated microlens arrays [60], and dynamic beam shaping control [61,62], will achieve tighter beam confinement and higher far-field collimation efficiency, thereby enabling long-range, low-power optical links.” (Paragraph 4, Section 3)

In the revised manuscript, we have added following references:

“57. Yu, L. et al. Adoption of large aperture chirped grating antennas in optical phase array for long distance ranging. *Opt. Express* **30**, 28112-28120 (2022).

58. Huang, Y., Yu, C., Chen, C. Sidelobe-suppressed silicon waveguide gratings through asymmetrically-apodized corrugations. *IEEE Photonics T.* **36**, 629-632 (2024).

59. Wang, Z. et al. Metasurface empowered lithium niobate optical phased array with an enlarged field of view. *Photonics Res.* **10**, B23-B29 (2022).

60. Im, C. et al. Hybrid integrated silicon nitride–polymer optical phased array for efficient light detection and ranging. *J. Lightwave Technol.* **39**, 4402-4409 (2021).
61. Zhao, S., Lian, D., Li, W., Chen, J., Dai, D. & Shi, Y. Low sidelobe silicon optical phased array with chebyshev amplitude distribution. *Nanophotonics* **13**, 263-269 (2024).
62. Sharma, A. et al. Optimization of a programmable $\lambda/2$ -pitch optical phased array. *Nanophotonics* **13**, 2241-2249 (2024).” (References)

References:

- [C17] Yu, L. et al. Adoption of large aperture chirped grating antennas in optical phase array for long distance ranging. *Opt. Express* **30**, 28112-28120 (2022).
- [C18] Huang, Y., Yu, C., Chen, C. Sidelobe-suppressed silicon waveguide gratings through asymmetrically-apodized corrugations. *IEEE Photonics T.* **36**, 629-632 (2024).
- [C19] Wang, Z. et al. Metasurface empowered lithium niobate optical phased array with an enlarged field of view. *Photonics Res.* **10**, B23-B29 (2022).
- [C20] Im, C. et al. Hybrid Integrated silicon nitride–polymer optical phased array for efficient light detection and ranging. *J. Lightwave Technol.* **39**, 4402-4409 (2021).
- [C21] Zhao, S., Lian, D., Li, W., Chen, J., Dai, D. & Shi, Y. Low sidelobe silicon optical phased array with chebyshev amplitude distribution. *Nanophotonics* **13**, 263-269 (2024).
- [C22] Sharma, A. et al. Optimization of a programmable $\lambda/2$ -pitch optical phased array. *Nanophotonics* **13**, 2241-2249 (2024).

Comment 5:

There is a typo in Fig. 4a, 'DCA' should be 'DAC'.

Reply and modifications:

We sincerely thank the reviewer for their meticulous review and for identifying the typographical error in Figure 4a. We have corrected “DCA” to “DAC” in the figure.

Figure 4. Experimental results of a high-speed OWC system utilizing the OPA. **a** Schematic of the 320 Gbps data transmission link.

To Reviewer #2's comments:

Recommendation: This manuscript presents a multi-target, ultra-high-speed optical wireless communication system based on a thin-film lithium niobate optical phased array. The system demonstrates impressive performance, including high-speed multi-target beam steering across a wide field of view, low sidelobe levels, and a record single-channel data rate of 320 Gbps using 16-QAM modulation. The work highlights significant advantages in integration, power efficiency, and dynamic reconfigurability, supported by comprehensive experimental data such as high-definition video transmission. The results are clearly presented and address a highly relevant research direction with considerable potential for next-generation wireless networks. The manuscript is well-structured, exhibits strong originality, and the data are systematically organized, providing robust support for the conclusions. This work is likely to attract broad interest from the photonics and communications communities. In my view, the paper merits publication after the authors address the following points:

Author Response:

We sincerely thank the reviewer for their positive assessment of our work and for raising these insightful points. We have carefully considered the comments and provide the following responses.

Comment 1:

The current vertical beam steering range is 11° . The authors should discuss potential strategies or design modifications to further expand the vertical field of view in future implementations.

Reply and modifications:

We thank the reviewer for this insightful suggestion. In the present design, the OPA achieves a vertical beam steering range of 11° . The steering efficiency is determined to be $-0.091^\circ/\text{nm}$ based on our experimental data. The current limitation in the vertical FOV is primarily due to the bandwidth constraints of passive devices and the wavelength tuning range of the external laser source.

Vertical steering angle control is achieved through wavelength tuning. The output coupling angle is determined by the grating equation and can be expressed as:

$$\sin \varphi = \frac{\Lambda n_{\text{eff}} - \lambda_0}{n_0 \Lambda}$$

where Λ is the antenna period, n_{eff} is the effective refractive index of the waveguide mode, λ_0 is the wavelength, and n_0 is the background refractive index. Therefore, for a fixed grating geometry, the achievable vertical field of view is directly proportional to both the laser tuning span and the effective refractive index difference.

To further expand the vertical steering range in future implementations, we propose three potential strategies. First, introducing a linearly chirped or apodized grating profile can broaden the effective radiation bandwidth, enabling a larger angular sweep over the same wavelength tuning range, as demonstrated in references [C1, C2]. Second, utilizing both TE and TM modes [C3, C4], made possible by the optical anisotropy of TFLN, can produce distinct vertical steering ranges, thereby significantly expanding the accessible FOV. Our simulations in Supplementary Material 10 confirm that this approach can at least double the FOV. Third, implementing a multi-period OPA design allows for the generation of diverse steering angles from gratings with differing periodicities, leading to an enlarged FOV, as supported by references [C5].

To provide additional clarity in the manuscript, we have included the following sentences in the revised Supplementary Information:

“The current vertical beam steering range of 11° is mainly limited by the spectral response of the bandwidth of passive devices and the wavelength tuning range of the external laser. Future designs could substantially extend this range by employing apodized or chirped gratings to increase radiation bandwidth ^[1,2], and exploiting polarization diversity ^[3,4] or multi-period ^[5] OPA in the thin-film lithium niobate (TFLN) platform. These strategies, combined with the integration of widely tunable on-chip lasers, are expected to expand the vertical FOV well beyond 20°, enabling more flexible beam steering for next-generation optical wireless communication (OWC) systems.” (Paragraph 4, Supplementary Note 3, Supplementary information)

In the revised Supplementary Information, we have added following references:

- [1] Yu, L. et al. Adoption of large aperture chirped grating antennas in optical phase array for long distance ranging. *Opt. Express* **30**, 28112-28120 (2022).
- [2] Huang, Y., Yu, C., Chen, C. Sidelobe-suppressed silicon waveguide gratings through asymmetrically-apodized corrugations. *IEEE Photonics T.* **36**, 629-632 (2024).
- [3] Zhao, S., Chen, J., Shi, Y. Dual polarization and bi-directional silicon-photonics optical phased array with large scanning range. *IEEE Photonics J.* **14**, 6620905 (2022).
- [4] Zhao, S. et al. Polarization multiplexing silicon photonic optical phased array with a wide scanning range. *Opt. Lett.* **48**, 6092-6095 (2023).
- [5] Zhang, L. et al. Large-scale integrated multi-lines optical phased array chip. *IEEE Photonics J.* **12**, 6601208 (2020).” (References)

References:

- [C1] Yu, L. et al. Adoption of large aperture chirped grating antennas in optical phase array for long distance ranging. *Opt. Express* **30**, 28112-28120 (2022).
- [C2] Huang, Y., Yu, C., Chen, C. Sidelobe-suppressed silicon waveguide gratings through asymmetrically-apodized corrugations. *IEEE Photonics T.* **36**, 629-632 (2024).
- [C3] Zhao, S., Chen, J., Shi, Y. Dual polarization and bi-directional silicon-photonics optical phased array with large scanning range. *IEEE Photonics J.* **14**, 6620905 (2022).
- [C4] Zhao, S. et al. Polarization multiplexing silicon photonic optical phased array with a wide scanning range. *Opt. Lett.* **48**, 6092-6095 (2023).
- [C5] Zhang, L. et al. Large-scale integrated multi-lines optical phased array chip. *IEEE Photonics J.* **12**, 6601208 (2020).

Comment 2:

The manuscript does not explicitly clarify the parameters governing the horizontal and vertical beam resolution, nor the ultimate factors limiting these resolutions. A

discussion on these aspects would help readers better understand the system's performance constraints.

Reply and modifications:

We thank the reviewer for raising this important point regarding the parameters governing horizontal and vertical beam resolution, and the factors limiting these resolutions. We appreciate the opportunity to clarify these aspects, as they are critical for understanding the performance boundaries of our OPA system.

The horizontal beam resolution, often characterized by its full width at half maximum (FWHM), is primarily determined by the overall aperture size, which is the product of the number of antenna elements (N) and their pitch (d). According to the relation $\theta_{\text{FWHM}} = \frac{0.886\lambda}{Nd \cos \theta}$, a larger aperture leads to a narrower beam, thereby improving the horizontal resolution. In our 16-channel OPA design with a 1.5 μm antenna pitch operating at a 1550 nm wavelength, the theoretical horizontal beamwidth is approximately 3.72°. This closely matches the measured FWHM of 3.8° shown in Figure 2c, confirming the consistency between experiment and theory.

The vertical beam resolution is mainly governed by the radiation characteristics of the trapezoidal grating antennas. The effective grating length defines the beam divergence in the vertical direction. In our device, the measured vertical FWHM is 1.72°, which aligns well with simulations based on the designed grating length.

To provide additional clarity in the manuscript, we have included the following sentences in the revised Supplementary Information:

“The horizontal and vertical resolutions of the OPA are governed by distinct parameters. The horizontal beam resolution, characterized by its full width at half maximum (FWHM), is primarily determined by the overall aperture size, which is the product of the number of antenna elements (N) and their pitch (d). According to the relation $\theta_{\text{FWHM}} = \frac{0.886\lambda}{Nd \cos \theta}$, a larger aperture yields a narrower beam, thus enhancing the horizontal resolution. Conversely, the vertical beam resolution is chiefly dictated by the radiation properties of the trapezoidal grating antennas. The effective grating length

sets the vertical beam divergence.” (Paragraph 6, Supplementary Note 2, Supplementary information)

Comment 3:

For multi-wavelength parallel beam control scenarios, could the authors comment on whether the SPGD algorithm requires separate objective functions for each wavelength? If a unified objective function is used, how is convergence stability maintained? Additionally, is sequential calibration necessary for the dual-wavelength operation demonstrated in the experiments?

Reply and modifications:

We sincerely thank the reviewer for raising this important question regarding multi-wavelength parallel beam control. In our experimental setup, we assign a separate objective function for each wavelength, defined based on diffraction efficiency, to address the specific requirements of individual wavelengths. To manage the complexity of the multi-wavelength optimization, we employ a weighted sum method that reduces the problem dimensionality effectively.

The convergence stability in the presence of multiple wavelengths is ensured by the physical properties of the OPA and the weighted optimization approach. Specifically, the phase shifters in the OPA control the beam direction, and when different wavelengths are incident simultaneously, the radiation direction for each wavelength shifts independently due to dispersion effects, as described in reference [C6]. This independence allows the beams to operate without mutual interference. We define an objective function for each wavelength and optimize all beams concurrently using a weighted method applied across the targets, which balances the optimization process and maintains convergence stability, as supported by prior work [C7].

For the dual-wavelength operation in our experiments, we do not use sequential calibration. Instead, we implement a parallel optimization strategy to avoid disrupting the voltage settings that might occur in a step-by-step approach. This ensures that the optimization for one wavelength does not compromise the results for others, leading to more robust performance.

[C6] Chen, J. et al. Single soliton microcomb combined with optical phased array for parallel FMCW LiDAR. *Nat. Commun.* **16**, 1056 (2025).

[C7] Marler, R.T., Arora, J.S. The weighted sum method for multi-objective optimization: new insights. *Struct. Multidisc. Optim.* **41**, 853–862 (2010).

Comment 4:

Prior to operation, does the SPGD algorithm require calibration at each beam steering angle? If so, could this initialization process limit the overall communication speed or target switching rate?

Reply and modifications:

We thank the reviewer for raising this pertinent question regarding the initialization process of the SPGD algorithm. The reviewer is correct to note that initial phase calibration is essential for high-resolution OPAs, primarily to compensate for inherent phase errors introduced during device fabrication.

In our system, once an initial calibration is performed at a reference angle (for instance, at the diffraction null), the obtained phase map effectively captures the static fabrication-derived errors. For any subsequent beam steering angle, the required phase pattern is generated by superimposing the ideal gradient phase for that angle onto the pre-calibrated error phase. Therefore, repeated full recalibration at each individual steering angle is not necessary.

While environmental perturbations can introduce minor dynamic phase shifts, our experiments confirm that the SPGD algorithm compensates for these efficiently, typically converging within only a few iterations. Furthermore, the final voltage configurations corresponding to various steering angles are stored in the memory of the host computer or FPGA. During operation, these pre-calculated values can be loaded onto the phase shifters almost instantaneously. As a result, the calibration process does not impose a bottleneck, and the overall communication speed and target switching rate remain unaffected.

Comment 5:

The discussion mentions a "transition time of approximately 158 ps", while the measured rise and fall times are 170 ps and 146 ps, respectively. The calculation or definition of the 158 ps value should be clarified for consistency.

Reply and modifications:

We thank the reviewer for highlighting this point. We agree that the original description of the transition time was not sufficiently clear, and we apologize for any confusion caused. The value of 158 ps is calculated as the arithmetic mean of the measured rise time (170 ps) and fall time (146 ps), which provides a representative estimate of the overall switching performance.

In the revised manuscript, we have clarified this point in the text as follows:

“The phase shifters exhibit measured rise and fall times of 170 ps and 146 ps, respectively, yielding an average transition time of 158 ps, and a modulation efficiency of 8.73 pJ/π.” (Paragraph 1, Section 3)

Comment 6:

On page 13, first paragraph, "tcoherent detection" appears to be a typographical error. This should be corrected to "coherent detection".

Reply and modifications:

We sincerely thank the reviewer for bringing this typographical error to our attention. The reviewer is correct that "tcoherent detection" is inadvertently misspelled and should read as "coherent detection."

In the revised manuscript, we have carefully corrected this error:

“On the receiver side, coherent detection is first applied.” (Paragraph 2, Section 2.3)

Comment 7:

Please verify the angle units in Figure 4d and ensure consistent capitalization of units in Table 1 (e.g., 'mW' vs. 'mw').

Reply and modifications:

We sincerely thank the reviewer for highlighting these important details and for their thorough review, which has significantly improved the quality of our manuscript. We have addressed both points as follows.

Regarding Figure 4d, we have carefully verified the angle units and confirm that all angles are correctly represented in degrees ($^{\circ}$). We apologize for any oversight in the initial presentation and have ensured that the units are clearly and consistently labeled.

In the revised manuscript, we have carefully corrected the errors:

“Figure 4. Experimental results of a high-speed OWC system utilizing the OPA. a Schematic of the 320 Gbps data transmission link. b DSP flowchart implemented at the transmitter and receiver. c Optical spectra of the 16-QAM signal recorded at different points within the system. d BERs measured at different wavelengths for transverse angles $\theta = 0^{\circ}$ and $\theta = 31^{\circ}$, respectively. e Recovered constellation diagrams corresponding to each angle.” (Figure 4, Section 2.3)

Table 1 Comparison of representative works in high-speed OWC.

Year	Method	On-chip	Dynamic beam scanning	FOV	SLS R	Single-channel speed	Switching speed	Modulation efficiency	Applications
2016 (ref. 42)	MEMS	Yes	Yes	4°	NA	10 Gbps (16 QAM)	NA	NA	Signal
2018 (ref. 43)	AWGR	No	No	$18.6^{\circ} \times 18.6^{\circ}$	NA	35 Gbps (OOK) 112 Gbps (PAM 4)	NA	NA	Signal/Video
2020 (ref. 44)	LCoS-SLM	No	Yes	6°	NA	60 Gbps (PAM 4)	NA	NA	Signal
2020 (ref. 45)	Metasurface & AWGR	No	No	35°	NA	20 Gbps (OOK)	NA	NA	Signal
2022 (ref. 46)	Metasurface	No	No	80°	NA	100 Gbps (OOK)	NA	NA	Signal
2023 (ref. 47)	Metasurface	No	No	$20^{\circ} \times 20^{\circ}$	NA	100 Gbps (QPSK)	NA	NA	Signal
2023 (ref. 48)	Metasurface & LCoS-SLM	No	Yes	$20^{\circ} \times 20^{\circ}$	NA	10 Gbps (OOK)	~ms	NA	Signal
2023 (ref. 49)	Metalens & Translation stage	No	No	80°	NA	10 Gbps (OOK)	NA	NA	Signal
2024 (ref. 11)	Metasurface	No	No	120°	NA	200 Gbps (OOK)	NA	NA	Signal/Video
2024	Topological	Yes	No	360°	NA	72 Gbps (16 QAM)	NA	NA	Signal/Video

(ref. 21)	beamformer								
2024 (ref. 9)	Silicon-based OPA	Yes	Yes	60°×14°	NA	12 Gbps (OOK)	24.4 μs	20 mW/π	Signal/Image
2024 (ref. 50)	Si ₃ N ₄ - Si-based OPA	Yes	Yes	96°×14.4°	-6 dB	NA	32.26 μs	1.33 W/π	Image
2025 (ref. 51)	Silicon-based OPA	Yes	Yes	25.6°×8.1°	-7.2 dB	40 Gbps (OOK)	NA	1.92 W (total)	Signal
This work	TFLN-based OPA	Yes	Yes	62°×11°	-13.6 dB	320 Gbps (16 QAM)	158 ps	8.73 pJ/π	Signal/Video

Note: NA: not measured, MEMS: micro-electrical mechanical system, AWGR: arrayed waveguide grating router, LCoS SLM: liquid crystal on silicon spatial light modulator.